Replication

cognition/psychology

inhibition of return, working memory, refreshing, attention

**Author for correspondence:**
Naomi Langerock
e-mail: naomi.langerock@unige.ch

# Inhibition-of-return-like effects in working memory? A preregistered replication study of Johnson *et al.* (2013)

Naomi Langerock[1], Giuliana Sposito[2], Caro Hautekiet[1] and Evie Vergauwe[1]

[1]Faculty of Psychology and Educational Sciences, University of Geneva, 40 Boulevard Pont d'Arve, 1211 Geneva 4, Switzerland
[2]School of Psychology, University of Padova, Via Venezia, 12/2, 35131 Padova, Italy

NL, 0000-0001-9582-6080; EV, 0000-0002-7339-2370

The present study concerns a preregistered replication of the study conducted by Johnson *et al.* (Johnson *et al.* 2013 *Psychol. Sci.* **24**, 1104–1112 (doi:10.1177/0956797612466414)), in which they showed an inhibition-of-return-like effect in working memory. Inhibition of return is a well-known phenomenon observed in the field of perception and refers to the observation that it takes longer to look back at a location which has recently been explored than to look at an unexplored location. Working memory is a central concept in the field of cognitive psychology and refers to the capacity to process and maintain information simultaneously over short periods of time. Johnson's study applied the inhibition of return paradigm to the concept of working memory. Their results showed that it is harder to access a working memory representation that had just been thought of, i.e. refreshed, in comparison to an unrefreshed working memory representation. Contrary to this study of Johnson *et al.*, who observed refreshing to result in inhibitory processes, most studies on refreshing have described its effect as increasing/prolonging the level of activation of the memory representations. In an attempt to integrate these opposite patterns produced by 'refreshing', we started by replicating one of the studies on the inhibition of return in working memory reported by Johnson *et al.*

## 1. Introduction

Inhibition of return is a well-known phenomenon in the field of perception. It refers to the observation that locations which have been explored visually are slower to be accessed just afterwards than locations that have not been visually explored just before.

**Figure 1.** Schematic representations of the inhibition of return paradigm used in perception (*a*) and in verbal working memory (*b*).

This observation was first reported by Posner & Cohen in 1984 [1]. In their basic paradigm (figure 1*a*), three boxes are shown on screen, horizontally aligned. At the start of the trial, one of the outer boxes is cued resulting in an automatic attentional shift to the cued box. Next, a target object is shown in one of the three boxes and participants have to press a key as soon as they detect the target object. Posner and Cohen found longer reaction times when the target was presented at the location that had been cued just before than when the target was presented in the box opposite to the cued box, even though the cued box has no predictive value regarding the location of the upcoming target object. Thus, they observed slower accessing of the already explored locations (i.e. inhibition of return). A short period of early facilitation (faster access) at the cued location was nevertheless observed when the probe followed the cue during the first interval of about 200 or 300 ms. When the probe was presented after this very short facilitative interval, an inhibitory effect was systematically observed. This observation concerns one of the most basic processes within the field of perception, i.e. orienting, and has hence boosted research of this phenomenon. The observation of inhibition of return in the field of perception, as well as its accompanying time course going from a short period of facilitation to a longer period of inhibition, have been replicated on many occasions [2–4], yet the underlying mechanism remains a subject of discussion [5,6].

Some years ago, Johnson *et al*. [7] published the article 'Foraging for thought: an inhibition-of-return-like effect resulting from directing attention within working memory', including two experiments that showed a pattern of results similar to the phenomenon of inhibition of return but in a working memory context. Working memory refers to the system capable of simultaneously processing and maintaining information over short periods of time. It is of crucial importance for our cognitive activities and is solicited almost continuously throughout our daily activities, like for example keeping in mind our shopping list while searching for these products in the supermarket, or keeping in mind

the beginning of a sentence while listening and processing what follows in that same sentence. One of the main processes being investigated within the field of working memory is 'refreshing', i.e. actively thinking back to information that is no longer perceptually present by focusing one's attention upon the internal representation of the information [8,9]. In terms of attentional processes, refreshing is hence concerned with internal attention (i.e. orienting towards information represented in working memory), while the classical inhibition of return effect is concerned with external attention (i.e. orienting towards information that is perceptually present in the surrounding environment; e.g. Chun [10], Kiyonaga et al. [11] or Verschooren et al. [12] for similar conceptions of internal versus external attention). Thus, by testing inhibition of return in a working memory context, Johnson et al. [7] were also testing the relationship between internal and external attention by testing whether key phenomena observed for external attention can also be found for internal attention.

The inhibition of return paradigm that was used by Johnson et al. [7] to test the effect in working memory can be seen in figure 1b. One can see the similarities with the classical inhibition of return paradigm used in the field of perception, shown in figure 1a. The temporal parameters of the original paradigm were slightly lengthened in the inhibition of return paradigm for working memory for the word-related task to be inserted. In Johnson et al.'s [7] experiment, participants were presented with two words displayed on-screen one above the other. After a short moment, these two words disappeared from the screen and were replaced by a cue, here an arrow pointing towards one of the two locations where the words had been presented. Participants were instructed to think back to, i.e. refresh, the word that was presented on that location before, and say out aloud this refreshed word. In the example here, the refreshed word would be the word 'cutlet'. Next, a single word was centrally presented on the screen, which participants had to read aloud. This word could correspond either to the refreshed word (cutlet), the unrefreshed word (bedlam) or a new word (rudder). Participants' response time to the onset of the last word reading was recorded and assumed to reflect the accessibility of the memory representation, with more accessible words displaying shorter response times and less accessible words displaying longer response times. Johnson et al. [7] observed that the words which had been refreshed (i.e. that had been the object of internal attention because they had actively been thought about) were slower to access immediately afterwards than the words that had not been refreshed just before, as reflected in longer response times for refreshed words than for unrefreshed words.

Johnson et al. [7] also report a second experiment showing this inhibition-of-return-like effect in working memory, this time in the visual domain instead of the verbal domain. In experiment 3 in Johnson et al. [7], participants were presented with two images, horizontally aligned, that could correspond to one of four different categories (chairs, faces, houses or shoes). Participants saw the two images on the screen, followed by an empty screen with the refreshing cue (i.e. think back and visualize), pointing towards one of the locations where an image had been shown previously. Then participants were presented with a probe image, centrally displayed but initially blurred by noise images that faded away progressively. As soon as participants recognized the category to which the probe image belonged, they were to press a stop button and then indicate this category. As for the experiment in the verbal domain, the probe item could correspond to the refreshed item, the unrefreshed item or a novel item. Results indicated that in the visual domain as well, participants were slower to access the refreshed item than the unrefreshed item, accumulating evidence for an inhibition-of-return-like effect in working memory.

These two experiments reported by Johnson et al. [7] suggest that refreshing in working memory leads to a slowing down of the accessibility of refreshed items, yet, this does not align with the general conception of refreshing in working memory. The concept of refreshing in working memory is generally described as a domain-general attentional mechanism which is used to prevent memory loss by boosting, prolonging and strengthening the activation level of remembered items [9]. There is ample evidence suggesting that refreshing of information in working memory leads to better memory performance. For example, Souza et al. [13] instructed participants to refresh coloured dots according to a number of refreshing cues and observed better memory performance for items that had been refreshed more often. Barrouillet et al. [14,15] have described a series of studies in which it is shown that increasing the free time during a working memory task results in better memory performance, and this free time is assumed to be used for refreshing activities. The process of refreshing as it has been described in the working memory literature so far is thus rather in terms of increased activation of memory items resulting in better memory performance, and not in terms of inhibitory processes.

While Johnson et al.'s [7] studies show inhibitory effects immediately after refreshing, they do show increased memory performance on a long-term memory test. In their experiment in the verbal domain (words), a surprise memory test was administered after about 20 min of performing another task

(experiment 1A). On this long-term memory test, better memory performance was observed for the refreshed items (independently of whether they had been the probed item afterwards or not), showing hence that refreshing boosts long-term memory performance. This is in line with previous studies which have shown that refreshing of memory items may boost memory performance not only for working memory but also for long-term memory [16–18]. The studies by Johnson et al. [7] showed hence a combination of inhibitory processes right after refreshing and increased recall performance in the long term.

A recent study, however, studied the immediate consequences of refreshing and found facilitation of the just-refreshed memory representations [19], instead of inhibition as was found by Johnson et al. [7]. Vergauwe et al. [19] were interested to see the direct impact of refreshing on the immediate accessibility of these refreshed items, rather than on the recall performance at the end of the trial as was the case in most of the aforementioned studies. In the paradigm of Vergauwe et al. [19], four letters were presented sequentially in four different boxes (going from upper left to lower right). After this initial presentation, a number of these four now empty boxes were highlighted in red in the same order in which the letters had appeared before (between one and five boxes were highlighted sequentially). Participants were instructed to refresh the letter that had been presented in the box as soon as it turned red. At the end of the trial, a probe letter appeared centrally, and participants had to indicate whether this letter corresponded to one of the four letters in the series presented (independently of whether it had been refreshed or not). This letter could correspond to the just-refreshed letter (i.e. the last letter that had been refreshed), the not-just-refreshed letter (i.e. a letter pertaining to the series of four letters, but not the last one that had been refreshed) or to a novel letter (i.e. a letter not pertaining to the series of four letters). Results showed that participants were faster to judge the probe when it corresponded to the just-refreshed letter than when it corresponded to a not-just-refreshed letter. This study thus found that refreshing resulted in items being more accessible and hence accentuates the facilitating effect of refreshing, contrasting with the inhibition-of-return-like effect observed by Johnson et al. [7] immediately after refreshing. In accordance with the results of this study by Vergauwe et al., a recent study [20] using a paradigm quite similar to the inhibition of return paradigm in working memory also observed facilitatory effects for the refreshed items. So despite the similarities in the paradigm used, no inhibition-of-return-like effects were observed in this study.

At this point, the inhibition-of-return-like effect in working memory described by Johnson et al. [7] does not seem to fit with the concept of refreshing as described in the working memory literature so far, which is rather in terms of increased activation instead of suppressed activation. Nevertheless, the process of refreshing has not revealed all of its characteristics yet and a deeper understanding of its functioning is necessary [9]. Knowing whether and how refreshing gives rise to an inhibition-of-return-like effect in working memory, together with observations of facilitatory effects of refreshing in working memory (both on immediate and delayed recall performance as on immediate response times), could reveal important knowledge on how information is maintained in working memory and the role of attention in working memory. Additionally, a better understanding of the inhibition-of-return-like effect in working memory may help us to understand the relationship between internal attention, operating on information that is no longer perceptually present, and external attention, operating on perceptually present information. Recent literature emphasizes the interactions and similarities between these two processes [10–12,21,22], and it may be theoretically relevant to gather additional knowledge about the boundaries of this comparison.

Returning to our primary aim regarding the immediate effect of refreshing on the accessibility of working memory representations, it is a possibility that temporal factors alter its effect, as Johnson et al. [7] had already suggested earlier and as is the case for inhibition of return in perception as well (going from facilitation to inhibition to neutral over time). For example, in the study by Vergauwe et al. [19], the probe appeared 1000 ms after the onset of the refreshing cue and this resulted in facilitated access to the working memory representation. In the paradigm used by Johnson et al. [7], the probe appeared 1600 ms after the onset of the refreshing cue and this led to inhibition to access the working memory representation. In a recent paper by Higgins et al. [23] using this same inhibition of return paradigm, inhibited access to the refreshed working memory presentation was confirmed when the probe appeared 1600 ms after the onset of the refreshing cue, while this inhibition seemed to disappear or at least decrease when the probe appeared 2000 ms after the onset of the refreshing cue.

Before launching our attempt to solve the discrepancy between facilitative and inhibitory effects of refreshing on the accessibility of working memory representations, we decided first to replicate at least one of the studies reported by Johnson et al. [7] showing the inhibition-of-return-like effect. Overall, the concept of refreshing in the working memory literature has always been linked to increased activation of working memory representations [8,9,13–15,24,25] and, to our knowledge, never to inhibition, except for the studies by Johnson et al. [7] and a follow-up by some of these same authors [23]. This inhibition-

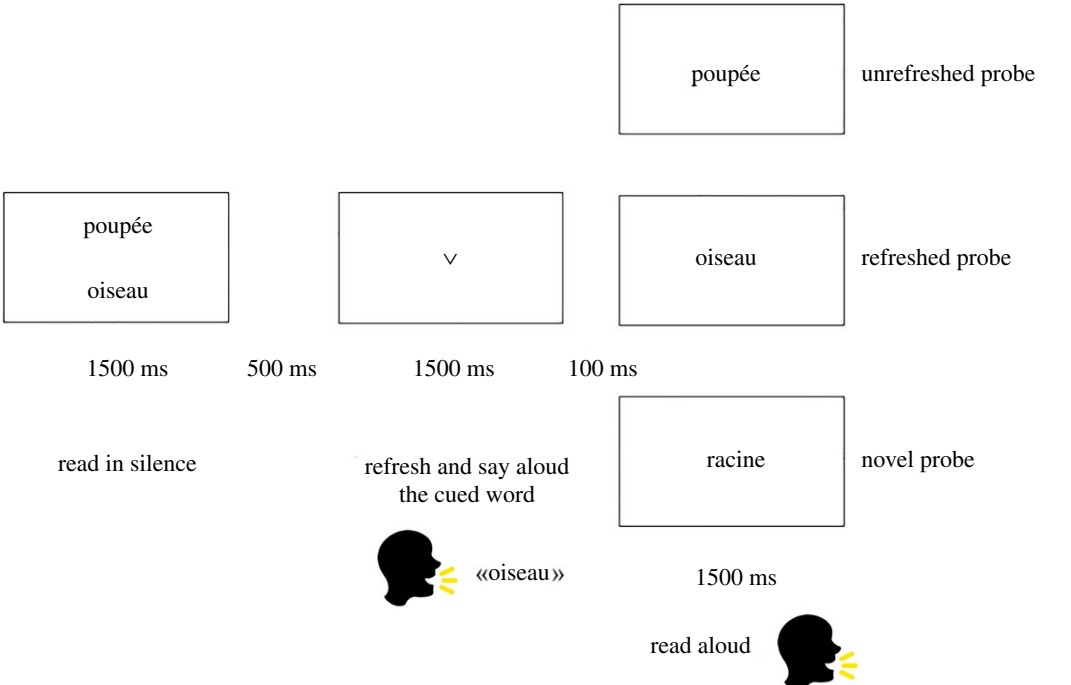

**Figure 2.** Example of a trial.

of-return-like effect asks hence for a replication before proceeding with a more detailed investigation of the immediate effects of refreshing on the accessibility of items represented in working memory.

The present study reports our replication of experiment 1B reported by Johnson *et al.* [7]. In this experiment, we test the hypothesis that words which have been refreshed just before are slower to access than words that have not been refreshed just before. The replication of this study was preregistered on aspredicted.org (https://aspredicted.org/s82zc.pdf) before the start of the data collection and the results and materials for this replication can be found on OSF (https://osf.io/3k6rv/). The method and analysis section of experiment 1B reported by Johnson *et al.* [7] were followed as closely as possible, although we tested the effect in a French-speaking population instead of an English-speaking population, and we used Bayesian sequential hypothesis testing instead of null hypothesis significance testing to examine if an inhibition-of-return-like effect is observed after refreshing in working memory. Bayes factors (BF) represent evidence in the data in favour or against a specific hypothesis, and evidence is considered to be moderate as from a BF of 3 on, in favour or against the hypothesis [26]. In the present experiment, the main hypothesis is tested by a one-sided Bayesian *t*-test because we have specific predictions about the direction of the hypothesized effect, and we aimed for a BF of 10 in favour of (or against) the hypothesis, which is considered strong evidence.

# 2. Methods

## 2.1. Participants

Johnson *et al.* [7] had a sample of 20 participants in their experiment 1B. We planned to start with 30 participants. We used Bayesian sequential hypothesis testing for our analysis and planned to continue increasing the number of participants by five until we obtained a BF of 10 for or against the hypothesis tested in our main *t*-test, or until we reached our predefined number of 60 participants, whichever comes first. In total, 35 participants were tested (29 female, mean age = 21.73 years). These were all undergraduate students at the University of Geneva, participating for course credit.

## 2.2. Materials and procedure

Participants performed the task as shown in figure 2. First, two words were presented for 1500 ms, one just above the centre of the screen, the other just below the centre of the screen. Participants were asked to read these words silently. A blank screen followed for 500 ms. Then, an arrow appeared on-screen for 1500 ms,

pointing to the location of one of the two previously presented words. The participants were to refresh and say aloud the word cued by the arrow while these words were no longer present on the screen. There then followed a blank screen for 100 ms and a final word appeared centrally on-screen for 1500 ms (i.e. the probe). Participants were asked to read aloud the probe as fast and accurately as possible. A microphone was used to register the onset of reading the probe by a voice key. The probe could correspond to (i) the cued word, i.e. refreshed-probe condition, (ii) the non-cued word, i.e. unrefreshed-probe condition, or (iii) a novel word, i.e. novel-probe condition. In total, there were 144 trials, with an equal distribution of the three probe types (48 of each). Trials followed each other with an inter-trial interval of 3000 ms.

Below, we further describe our implementation of the experiment. In particular, we describe in turn: (i) a selection of word stimuli for our French-speaking population, (ii) methodological details of our experiment that were not mentioned in the method section of the original paper, and (iii) methodological details that deviated from the original method.

### 2.2.1. Selection of word stimuli for our French-speaking population

Whereas the original experiment was run in an English-speaking population, our population was French-speaking. Therefore, we created a new set of 336 words. Not much information was given about the creation of the wordlists in the original paper except that 'stimulus lists' were equated for word length, frequency, number of phonemes, number of syllables and average time to read and that 'all conditions and lists were fully counterbalanced across participants' [7, p. 1105]. We proceeded as follows to implement this with the French stimuli. A set of nouns within the Lexique 3 database [27] was selected, all consisting of six letters, two syllables, four or five phonemes, and a frequency (book frequency) between 2 and 50. This resulted in a list of 747 words. Plural nouns were taken out, as well as ambiguous words, words with an obvious negative connotation, words with double significations, words derived from English words, words pronounced the same as another word in the word list and derivations of verbs. This left us with 396 words, of which the 336 most frequent words were selected for our experiment. Of the remaining 60 words, the most frequent 16 words were used for the training phase. Compared to Johnson *et al.* [7], the only criterion not considered for our word list was the reading time, as these were not available for the French words we had selected. Owing to the counterbalanced design of the study, these reading times were to be equally distributed across conditions anyhow. Next, we created seven lists of 48 words (list A, list B, …, list G). These seven lists were equal in terms of frequency ($BF_{01} = 375$) and number of phonemes ($BF_{01} = 53$). Fourteen different versions of the experiment were created as follows. The first list of words (i.e. list A), ordered by frequency (highest to lowest frequency), served as the novel probes in the first version of the experiment. The words in the remaining six wordlists (i.e. list B, list C, list D, list E, list F and list G) were collapsed in a general wordlist (containing 288 words) and ordered by frequency (highest to lowest) as well. These words, ordered from highest to lowest frequency, served as the word pairs presented at the beginning of each trial and hence as the refreshed and unrefreshed probe as well. Words from the novel as well as from the general wordlist were selected sequentially thus always showing the words highest in frequency at the beginning of the experiment and the words lowest in frequency near the end of the experiment. This way, it was reassured that the response time to read the probes was not affected by differences in frequency of the refreshed, unrefreshed or novel probe. Additionally, the use of fixed lists allowed the experimenter to have a printed list of the refreshed words and read aloud words, which drastically facilitates scoring the accuracy of these words during the experiment.

We repeated the process for each list. This resulted in seven primary versions of the experiment, whereas each list served once as the novel probes. Then, for each primary version, we created a parallel version, simply reversing the position in which the words of each pair appeared, i.e. a word appearing on the first trial as the word above the centre was now presented as the word below the centre and vice versa. This led to a total of 14 different versions of the experiment, that were counterbalanced over the participants, i.e. with the word list used for the novel probes rotating and the order of the word presented above or below the centre of the screen inversed for every other participant.

### 2.2.2. Methodological details of our experiment that were not mentioned in the method section of the original paper

In Johnson *et al.* [7], no details were reported regarding the presence and nature of a training phase in the original study. We decided to add a training phase in which the task was explained to the participants, followed by six training trials (two of each probe type, one for each arrow direction).

Furthermore, the original paper did not contain any information about the order of the 144 trials nor about the distribution of the direction of the refreshing arrows. In our experiment, trials with different

probe types (refreshed probe, unrefreshed probe or novel probe) were presented in a completely random order, with a break every 48 trials (i.e. two breaks in total). The order of the pointing direction of the arrow was the same for all participants and based on a randomization defined beforehand (each arrow direction occurring 72 times during the experiment). This predetermination allowed both words of each word pair to be refreshed in at least one version of the experiment. The randomization defined beforehand also allowed the experimenter to verify directly whether the participants spoke out aloud the correct to be refreshed word, as a response sheet for the refreshed word was prepared for each participant. As the different probe types were randomly attributed on every trial, there was no confound between probe types and the arrow direction.

In our experiment, participants sat at a comfortable distance from the screen (approximately 60 cm). The experimenter sat next to the participants during the experiment and registered: (i) whether the participant spoke out aloud the correct to be refreshed word, (ii) whether the participant read the probe word correctly, and (iii) whether no supplementary noise was present during the trial (e.g. coughing or other noises that interfered with the correct detection of the word reading onset time by the voice key). The experiment was run on a computer using E-prime 2.0, with the E-prime response box and a microphone connected in order to register the response onset time to read the probe.

### 2.2.3. Methodological details that deviated from the original method

The only explicit deviation from the original method concerns the recording of the response times to read the probes. While Johnson *et al.* [7] used a custom MATLAB script to detect sound exceeding a certain threshold with the option to manually adjust, we used a microphone connected to the E-prime response box recording the onset of sounds exceeding a certain threshold. We did not proceed to manually adjust these response times but directly excluded all trials on which supplementary sounds were detected by the experimenter because these sounds could interfere with the recording of the response time to read the probe. As such, our trial-by-trial exclusion criteria may have been slightly stricter than the ones used in the original paper and additional, preregistered, exclusion criteria were used.

## 2.3. Exclusion criteria

We used similar but slightly stricter exclusion criteria than those reported in the original study. Our preregistered exclusion criteria were as follows (https://aspredicted.org/s82zc.pdf). In both Johnson *et al.* [7] and the present study, trials were excluded in which participants stammered, misspoke or spoke too quietly for their response to be detected. In the present study, we applied these criteria when participants had to read the probe word as well as when they had to say aloud the refreshed word. We assumed this was the case in the original paper as well, though this was not explicitly stated. While in the original study manual adjustments were made on some occasions regarding the response times, we did not proceed in this way and excluded the trial as soon as there was a doubt regarding the correct recording of the response time. Additionally, we excluded trials on which the response time to read the probe word was below 150 ms, as we considered this may rather concern delayed speech related to saying aloud the to be refreshed word and not speech related to reading the probe word. As in the original paper, only valid trials were considered for analysis. As an additional criterion in our study, the datasets of participants with less than 75% of valid trials were discarded, in order to exclude participants not performing the task properly or participants with a limited understanding of the words used in the task if French was not their first language. Two participants had to be excluded as they did not reach the 75% criterion, and for the remaining participants, 6.4% of the trials had to be excluded (6.0%, 7.3% and 5.8% for the refreshed probe, the unrefreshed probe and the novel probe conditions, respectively). Of these invalid trials, 67% were owing to the absence of recording through the voice key.

## 2.4. Planned, preregistered analyses[1]

We followed the analysis as proposed in the original study but using Bayesian statistics. First, a Bayesian one-way repeated measures ANOVA was run on the response times of the valid trials, with probe type as within-subjects variable containing three levels (refreshed-probe condition, unrefreshed-probe condition

---

[1]***Additional exploratory analyses:*** in case, our study would not replicate the inhibition-of-return-like effect observed by Johnson *et al.* [7], a series of additional exploratory analyses would be run to find out where this discrepancy comes from, including analyses that explore potential differences between the English and French stimuli.

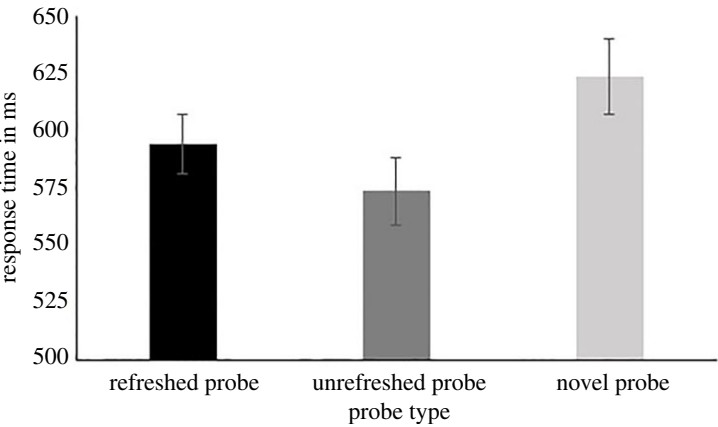

**Figure 3.** Response time as a function of probe type, errors bars represent the s.e.m.

and novel-probe condition), using JASP with default settings. Next, two one-sided Bayesian $t$-tests were run using JASP with default settings: (i) response time novel probes > response time old probes (whether refreshed or unrefreshed), and (ii) response time refreshed probes > response unrefreshed probes. This latter $t$-test concerned our main hypothesis.

This article received results-blind in-principle acceptance (IPA) at *Royal Society Open Science*. Following IPA, the accepted stage 1 version of the manuscript, not including results and discussion, was preregistered on the Open Science Framework (OSF) (https://osf.io/59rjk). This preregistration on OSF was performed after data analysis. Yet as stated before, a preregistration of the hypothesis, stop rule, exclusion criteria and planned data analysis had been done before starting the data collection on aspredicted.org (https://aspredicted.org/s82zc.pdf).

## 3. Results

As planned, 30 participants were tested in a first wave. Two participants had to be excluded as they did not reach the predetermined criterion of 75% valid trials. The analyses were performed on the response times of valid trials of the remaining 28 participants. The one-way repeated measures ANOVA on the response times, taking into account the three different probe types (refreshed, unrefreshed and novel) resulted in very strong evidence in favour of an effect of probe type ($BF_{10} = 799082$). The first one-sided $t$-test showed very strong evidence that novel probes were accessed more slowly than old probes ($BF_{10} = 31326$). Generally, BFs exceeding 10 are considered strong evidence, and exceeding 100, as is the case here, as extreme evidence [26]. Concerning our main hypothesis, whether refreshed probes were accessed more slowly than unrefreshed probes, the second one-sided $t$-test showed some support for inhibition of return ($BF_{10} = 2.54$), although BFs below 3 are generally considered inconclusive [26].

Following our preregistration, we added five participants because the BF of our main $t$-test was below 10, suggesting some, yet not conclusive, evidence. These five participants all had more than 75% valid trials, leaving us with a total of 33 participants to be included in the analysis. The one-way repeated measure ANOVA on the response times resulted again in very strong evidence for an effect of probe type ($BF_{10} = 1.99 \times 10^7$). The first one-sided $t$-test showed again abundant evidence that novel probes were accessed more slowly than old probes ($BF_{10} = 538083$; 624 ms versus 584 ms). Finally, concerning our main hypothesis, whether refreshed probes were accessed slower than unrefreshed probes, we now found strong confirmatory support for inhibition of return ($BF_{10} = 28$; 594 ms versus 574 ms; figure 3). Thus, the present study replicates the findings of Johnson *et al.* [7]: working memory representations that have just been refreshed are more slowly accessible right afterwards.

The preregistered hypotheses were confirmed by the preregistered planned analyses. The proposed exploratory analyses to investigate possible differences in the results between this replication study and the original study were hence not necessary.

## 4. Discussion

The main goal of the present study was to attempt to replicate the inhibition-of-return-like effect observed in the domain of working memory by Johnson *et al.* [7]. In their study, they observed that

working memory representations that had just been refreshed were accessed more slowly right after this refreshing. The results of the present replication study replicate and confirm this inhibition-of-return-like effect in working memory.

We tried to follow the experimental procedure of Johnson *et al.* [7] as closely as possible. The main difference concerned the words used, as the original experiment took place in an English-speaking population, whereas our replication was done in a French-speaking population. Furthermore, different software was used to run the experiment and to allow the voice key registration, Bayesian statistics were used instead of frequentist statistics, and some additional exclusion criteria were applied. The paradigm and the temporal parameters were nevertheless matched meticulously between the two studies. Proceeding this way, we replicated the results obtained by Johnson *et al.* [7].

Concentrating on the main analysis, which compared the response times of reading refreshed versus unrefreshed probes, Johnson *et al.*'s [7] participants took about 463 ms to start reading the refreshed probe and about 438 ms for the unrefreshed probe. The difference in response time between the two probe types was hence 25 ms, in favour of the unrefreshed probe. In the present replication study, participants took about 594 ms to start reading the refreshed probe and about 574 ms for the unrefreshed probe, resulting in a difference of 20 ms between the two probe types, in favour of the unrefreshed probe. Thus, even though the participants in the present study were slightly slower to respond to the probes in general (this was also the case for the novel probes), the difference in response times to respond to the refreshed versus unrefreshed probes was very similar between the original study and our replication. Our observed response time difference is also similar to that observed in experiment 1A of Johnson *et al.* [7], i.e. 23 ms, which used the same methodology but adding a long-term memory test, as well as to the response time difference in the experiment reported by Higgins *et al.* [23], in the condition testing young adults with the same temporal parameters, i.e. 13 ms. While the difference in response times between the refreshed and unrefreshed probes is numerically not that large, statistical analyses show it to be consistent, now also across laboratories and languages. As such, by replicating the inhibition-of return-like effect previously observed by Johnson *et al.*, our results clearly diverge from other studies that have found an immediate facilitative effect of refreshing, as had been observed in the studies by Vergauwe *et al.* [19] and Lintz *et al.* [20]. At the moment, several hypotheses as to why these results diverge could be proposed, based on methodological differences between these different studies. We will at this point refrain from speculating about these hypotheses and leave it up to future studies to clarify what is at the origin of this divergence and how these two effects (inhibition and facilitation) can be integrated into a comprehensive account of refreshing.

## 5. Conclusion

The current replication study shows that the inhibition-of-return-like effect in working memory can reliably be observed across different laboratories and languages when using this exact same paradigm. Knowing this, we can start thinking about the implications this has for our understanding of the process of refreshing and the operation of the focus of attention, as well as regarding the similarities between external and internal attentional processes.

Ethics. This study was approved by the ethical commission of the Faculty of psychology and educational sciences of the University of Geneva. All participants signed an informed consent.
Data accessibility. All data and material are available on OSF (https://osf.io/3k6rv/).
Authors' contributions. N.L.: substantial contributions to conception and design, analysis and interpretation of data, drafting the article or revising it critically for important intellectual content, final approval of the version to be published, agreement to be accountable for all aspects of the work in ensuring that questions related to the accuracy or integrity of any part of the work are appropriately investigated and resolved. G.S.: substantial contributions to data acquisition and interpretation of data, drafting the article or revising it critically for important intellectual content, final approval of the version to be published, agreement to be accountable for all aspects of the work in ensuring that questions related to the accuracy or integrity of any part of the work are appropriately investigated and resolved. C.H.: substantial contributions to analysis and interpretation of data, drafting the article or revising it critically for important intellectual content, final approval of the version to be published, agreement to be accountable for all aspects of the work in ensuring that questions related to the accuracy or integrity of any part of the work are appropriately investigated and resolved. E.V.: substantial contributions to conception and design, analysis and interpretation of data, drafting the article or revising it critically for important intellectual content, final approval of the version to be published, agreement to be accountable for all aspects of the work in

ensuring that questions related to the accuracy or integrity of any part of the work are appropriately investigated and resolved.

Competing interests. We declare we have no competing interests.

Funding. This research was conducted with support to E.V. and N.L. from the Swiss National Science Foundation (grant no. PCEFP1_181141 to E.V.).

Acknowledgements. We thank Mathilde François for helping with the data collection.

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
