## [Peer Review File · Royal Society Open Science]

Review History

RSOS-210254.R0 (Original submission)

Review form: Reviewer 1 (Colin McCormick)

Do you have any ethical concerns with this paper?

No

Have you any concerns about statistical analyses in this paper?

No

Recommendation?

Accept with minor revision

Comments to the Author(s)

Background/Summary:

- In a 2013 study, Johnson *et al.* published a 4-experiment paper which shows that participants are slower to respond to stimuli that were 'refreshed' (a sort-of 'cued reflection') immediately before, compared to stimuli that were not refreshed, but were previously presented.

An analogy was made with IOR, an attentional effect in which participants are slower to respond to/fixate on previously-disengaged locations.

- Some participants also completed a memory task, which showed that refreshed-probe trials have improved recall, adding a layer of intrigue to this effect.
- Because this runs contrary to the understanding of how refreshing increases activation, instead of suppresses (re: Johnson), and because this has not been supported by other research (to their knowledge), the current authors look to replicate one of the experiments presented.

Disposition

I believe this is an important and relevant replication to run, especially before using resources to try to understand the time-course/other properties of this 'IOR-like' effect. I only have a few concerns, which I am confident the authors will be able to effectively address.

Comments and Concerns

- I think you did a great job of outlining your statistics, and additionally how you plan on achieving appropriate power while collecting data.
- The methods are effectively the same, and the authors do a good job justifying any minor differences between the studies.
- I feel as if the authors could have used more citations to support some of the claims being made throughout the introduction. A couple examples:
 - o "One of the main processes being investigated within the field of working memory is "refreshing", i.e., actively thinking back to information that is no longer perceptually present by focusing one's attention upon the internal representation of the information. In terms of attentional processes, refreshing is hence concerned with internal attention (i.e., orienting towards information represented in working memory), while the classical inhibition of return effect is concerned with external attention (i.e., orienting towards information that is perceptually present in the surrounding environment)." I think this may need a few citations.
 - o "Overall, the concept of refreshing in the working memory literature has always been linked to increased activation of working memory representations and, to our knowledge, never to inhibition, except for the studies by Johnson et al. and a follow-up by some of these same authors." I would like to see some of the literature cited earlier in the paper cited again here to support this claim, but I also think that maybe more studies/information could be included in support of activation
- "Yet, studies showing increased recall performance after refreshing do not necessarily contradict Johnson et al's results..." I don't think that this is worded as accurately as it can be, as there is explicitly no contradiction. Jonson et al. showed improved recall for refreshing, so it is congruent with the literature in this aspect. I think this interesting discrepancy between this IOR-like effect and the LTM improvement for refreshed probes is worth discussing, but it must be done effectively to not confuse/mislead the reader.
- While I generally understand what will constitute a successful replication based on what you have written (you do talk about your 't-test' and such), I think you could more explicitly outline what your hypothesis is, and what exact statistical outcomes will support that hypothesis.

Review form: Reviewer 2 (Matthew Johnson)

Do you have any ethical concerns with this paper?

No

Have you any concerns about statistical analyses in this paper?

No

Recommendation?

Accept in principle

Comments to the Author(s)

To start off, full disclosure: I am Matt Johnson, the first author of the paper being replicated here. I reviewed this manuscript with the help of two of my Ph.D. students, Evan Lintz and Zach Cole. It should be noted that we had some prior familiarity with this line of work and had previously chatted with the authors about it at a conference. Our two research groups have never actually collaborated, so I don't think this is any kind of conflict of interest, but the context may be helpful.

For obvious reasons, we may be a bit biased, but we think generally this line of work is interesting and deserving of replication/extension, so overall we are on board with this effort. We have a few suggestions, questions, and comments, although none of the issues we'll be bringing up are serious enough to be considered deal-breakers for ultimate publication... we are merely trying to do what we can to make the ultimate published paper the best it can be. In no particular order:

1) A general note: There are a number of places where the authors note (fairly) the lack of certain methodological details in the original paper being replicated. We are certainly aware of those omissions... unfortunately, due to the very stringent word count limits of Psychological Science and the numerous requests of the reviewers at the time, we had to be extremely terse throughout the paper to fit everything in, and that required cutting out some of the finer details of the methods during the revision process. Some of these added details can be found in Higgins et al (2020), which the authors cited and which mostly uses the same methods as the original 2013 paper, but even Higgins et al omitted some of them. Thus, we would be happy to provide additional methodological details if the authors feel like including them for comparison would be useful. Of course normally back-channel communication between authors and reviewers is frowned upon, but in this instance since open review is encouraged anyway, I assume it would probably be OK to send an email or two back and forth to request/deliver any desired details. (If we really want to be on the up-and-up, I guess we could disclose the contents of those messages to the editors, just in case there are any concerns.) Anyway, if the authors prefer to restrict their discussion to the details that have been given in the official peer-reviewed version of the paper, that is fine with us too; we just wanted to make the offer in case it would help improve this paper and/or inform the authors' future work.

(N.b.: Hereafter when we mention page numbers, we'll use the authors' original page numbers, not the page numbers imposed by the manuscript submission system. I.e. when we say page 4, we mean the authors' original page 4, which is labeled "page 6 of 15" in the RSOS header and is actually the 7th page of the PDF file.)

2) We found the framing of some parts of the introduction just a tiny bit odd. For example, in lines 39-44 of the Summary ("Contrary to this study..."), as well as in lines 22-26 on page 4 ("While these two experiments...") and in lines 39-42 on page 5, the authors seem to be setting up the original finding as a bit of a paradox -- why did Johnson et al (2013) seemingly find inhibition when others find facilitation from refreshing? If one accepts the framework our group tends to use in which mental attention is assumed to have many of the same attributes as perceptual attention, it really is not that paradoxical... just like perceptual attention, it seems reasonable that mental attention could involve both short-term and long-term facilitation, but have a brief "refractory period" after the initial facilitation; hence the original comparison to perceptual IOR. (And of course such refractory periods are a fundamental part of how the brain operates, as we

can see in both the action potential and in a standard hemodynamic response in fMRI, although we admit those do not necessarily entail that every excitatory mental process must necessarily be followed by a behaviorally observable period of inhibition.) Anyway, this point is not too big of a deal... we just thought that this framing set up the original effect as perhaps a bigger mystery than it really was. (Granted, it surprised us a little bit when we initially found the effect -- our behind-the-scenes story is that we also expected these experiments to produce short-term facilitation and had intended to follow them up with an entirely different line of experiments -- but we came around to the IOR-like viewpoint fairly quickly after that initial surprise. We also do admit that there are some remaining questions around why some study designs show these IOR-like effects and relatively similar designs show facilitation, and we are actually doing some of those experiments ourselves right now.)

3) Again, not particularly necessary, but in the discussion of refreshing producing better memory performance (lines 22-45 on page 4), the authors may want to also mention the work of Marcia Johnson et al (e.g. Johnson et al 2002 -- "Second thoughts versus second looks: An age-related deficit in reflectively refreshing just-active information"), which as far as I'm aware is the earliest example of such findings (at least with designs in which the task/process is called "refreshing" -- similar designs might have existed earlier in cognitive psychology history under different names, but if so I am not aware of them).

4) Once more, not particularly necessary, but we have an in-press paper that might be relevant to the discussion around pages 4-5 of previous facilitative refresh effects. Lintz & Johnson (in press at *Cognition*, Exp 2) also found short-term facilitation for refreshed words in a design fairly close to designs we have previously used that produced IOR-like effects. This is by no means an ego trip and the authors should not feel compelled to cite that paper, but you may find a comparison of that experimental design to other designs useful in trying to work out what are the critical factors that differentiate IOR-like-effect-producing experimental designs from facilitation-producing designs. A preprint is available at <https://psyarxiv.com/z52vf/> if it is not available at *Cognition* in time for you to read it there.

5) Very minor -- typo on line 36 of page 7 -- "The probe could respond to..." should, I think, be "The probe could correspond to..."

6) As noted in point #1, we are happy to provide more detailed methodological information from our original study if the authors want it. However, the good news is that in many respects the authors did more-or-less similar things to our original methods anyway. Some highlights from memory (which as we all know is unreliable -- but I think these details are all correct): We used nouns of 1 or 2 syllables that were not exactly constrained to have all six letters, but which averaged around that same length (I would have to look up the exact number, but am happy to do so if it's wanted). We then also tried to remove highly valenced words and duplicates, although not in exactly the same manner as in this current study, but with similar intentions and (we expect) similar results. For future reference, in case the authors ever want to use it, we balanced our word lists with a tool that we have now improved and made public as the LIBRA toolbox in Matlab (manuscript under review, preprint at <https://psyarxiv.com/64yfw/>). We also used a short training phase which, if I recall correctly, also had six trials. Similarly, we also split the 144 trials into three blocks of 48 with breaks in-between. We also ran the original study on E-Prime, although at the time I believe we were still on version 1. In addition to digital recordings and the Matlab script we used, we actually also used an E-Prime voice key apparatus similar to the authors -- we in fact did do analyses with both the digital recording/Matlab script and with the E-Prime voice key numbers, and found them to largely agree... we just found that the digital recordings gave more precise results (less variance), and those were always intended to be the "primary" measure with the E-Prime as a backup, so with the space constraints of the original article, we only published the digital recording results. (If the authors ever want to do

analyses with digital recordings, the script is not yet public but it is fairly user-friendly after a bit of introduction, so we'd be happy to share that too for future work.) Our exclusion criteria were also pretty similar: Although the digital process we used allowed us to keep a few more trials with extraneous noise in them that did not directly involve the task (e.g. noises from inside or outside the testing room from someone moving around, accidentally bumping the microphone, doors closing, etc.), we generally excluded any trials where the noise was during the cue or probe period when speech was supposed to be occurring, including subject-generated noises like sniffs or throat clearing. (However, when the only noise was a small popping sound from the subject's mouth opening, we left the trials in but made sure the detected onset was for the first recognizable phoneme, not the pop of the mouth opening.) The authors are correct in assuming that if either the response during the refresh cue OR the response during the probe was excluded, we excluded the entire trial from analysis. (However, we only reported RTs for the probe period, again due to space constraints; if we had reported RTs for refreshing, I think we would have left in refresh instances where the subject subsequently misspoke on the probe, because the refresh precedes the probe temporally.) I suppose you could say our exclusion criteria for rejecting too-early responses was similar, although since we had the digital recording it was easy to listen back and ascertain the reason for the early trigger; since it is effectively impossible to have a legitimate response onset occur in >150ms, any such instances were either accidental speech sounds (in which case the trial was excluded) or incidental noises like mouth pops or environmental noise (in which case the trial was included, as long as the noise did not appear to interfere with either the subject's generation of a correct response or our ability to resolve it in the recording). Anyway, this has gone a bit long (although there are plenty more details that have been omitted, but which we'd be happy to provide) -- but all in all, I'm not 100% sure it's fair to say that "exclusion criteria were slightly stricter than the ones used in the original paper" (page 10, lines 31-33)... although it is reasonable to see how that conclusion could be reached, given the limited amount of detail we had room to include in the Psych Science paper. It might be fairer to say that the E-Prime voice key vs. the digital recording method simply allow different criteria to be used, because they are fundamentally different in character. With all that said, I think the choices made in the current study using the E-Prime voice key are reasonable and very similar to the criteria we would have used if we didn't have digital recording data ourselves.

7) We mostly found the counterbalancing scheme used here reasonable (see next point though), although it was different from the one we originally used. Briefly, from memory (we can give more details and/or check the veracity of these memories if desired), we effectively generated nine balanced word lists and used a 3x3 counterbalancing scheme, wherein we rotated through conditions (e.g., relative to counterbalance version 1, on counterbalance version 2 all refreshed-probe trials become unrefreshed-probe trials, unrefreshed-probe become novel-probe, and novel-probe become refreshed probe) and also word positions (e.g., relative to counterbalance version 1, on counterbalance version 4 all the initially-top words become initially-bottom words, initially-bottom become novel-position words, and novel-position words become initially-top words). This also leads to some oddities insofar as in each version, 2/9 word lists are not seen (the words ostensibly in the novel-position but on refreshed-probe or unrefreshed-probe trials, where the novel word is "invisible"), but as far as we could tell, it was methodologically sound.

8) For the counterbalancing scheme actually used here -- as noted, it seems mostly reasonable to us, but we did find it slightly odd that the authors chose to order the words from highest to lowest frequency instead of going with some kind of randomization/pseudo-randomization scheme that would still control for frequency effects (at least, on average) but without a potentially noticeable increase in word rarity throughout the duration of the experiment. As far as we can tell, this does not cause any actual problems for the interpretation of any results, it's just kind of an unusual choice for this kind of experiment... and if the results are the same as what we remember seeing in preliminary form at a previous conference, it doesn't seem to have caused any issues for successfully replicating the original work. Still, it might be worthy of some

additional discussion in the Discussion section of the final paper... and it might be an interesting question in itself to compare effects between block 1 and block 3, to see if the frequency manipulation had any noticeable consequences. We are not requiring that or anything, though; it just might be interesting to check. (Of course, then frequency would also be confounded with the simple passage of time, so it would be hard to say anything definitive... but if there are any hints of something interesting, we could probably help out by doing the same analysis in our original data, to save the authors some effort if they are thinking of pursuing any follow-ups on that question.)

9) The choice of one-sided Bayesian t-tests momentarily caused us to raise an eyebrow, given how one-sided tests are generally frowned upon in the NHST domain these days, even for situations in which they might have been historically considered an appropriate choice. However, after doing a bit of background reading, we have convinced ourselves that one-sided tests don't really deserve the same kind of stigma in the Bayesian domain, so we don't consider it an issue. Still, the authors may want to consider including a bit more justification (just a sentence or so, probably) in order to provide similar assurance to other readers who are not too familiar with Bayesian tests and their interpretations.

I think that's everything we found worthy of comment -- again, we generally support this line of work and are mostly providing those notes for the sake of completeness and for the authors' consideration in preparing the next version of the manuscript; none of them are anything we view as super-critical to the ultimate value of the work or the viability of the final paper. We'll be looking forward to seeing the results, and again, just let us know if you'd like us to provide any other methodological background on the work being replicated.

Best wishes,
Matt Johnson, Zach Cole, and Evan Lintz

Review form: Reviewer 3

Do you have any ethical concerns with this paper?

No

Have you any concerns about statistical analyses in this paper?

Yes

Recommendation?

Accept with minor revision

Comments to the Author(s)

The paper is generally well written and clear. I have only some minor suggestions to clarify a few points and extend the data-analysis to address some potential confounds.

First, it is important to unambiguously state that the inhibition of return occurs in a paradigm in which the cue is non predictive of the upcoming location of the target. When they are predictive facilitation is generally observed. The way it is written in the text, this feature is not mentioned which makes it seem as if attended locations are inhibited in perception. But this general statement is not true. The effect is observed because of the sudden onset capture caused by the box flashing on the screen.

Second, Figure 1B should present the words in English. After all, it is meant to describe the study of Johnson et al. which was with English speaking participants. The authors have an additional figure for their methods which can illustrate the procedure with French words. The figures could also include timings used for maximal information.

Third, the authors only propose to perform a t-test on reading times. I believe an additional analysis should also be reported that takes condition as a fixed effect and item as a random effect in a mixed effects model. Given the uncertainty regarding reading times for the French words, it is important to make sure that the results are not confounded by some outlier words. Reading times during the refreshing phase could also be analyzed to provide a general baseline of spoken times for the words.

Fourth, the authors should consider whether baseline reading times differ between their study and the ones previously reported in the literature that they are aiming to replicate. This may deserve a brief discussion.

Fifth, I was not sure about this exclusion of the trials with reading times below 150 ms. Wouldn't this affect only previously refreshed trials? because in neutral and unrefreshed trials, there would be no way a delayed speech onset from the refreshing phase could lead to a correct response for analysis. To alleviate these concerns, an analysis including and excluding these trials should be provided, as well as specific information about how many trials were excluded because of this in each design cell.

Decision letter (RSOS-210254.R0)

Dear Dr langerock

On behalf of the Editors, I am pleased to inform you that your Manuscript RSOS-210254 entitled "Inhibition-of-return-like effects in working memory? A preregistered replication study of Johnson et al. (2013)." deemed suitable for in-principle acceptance in Royal Society Open Science subject to minor revision in accordance with the referee and editor suggestions. Please find their comments at the end of this email.

The reviewers and handling editors have recommended publication, but also suggest some minor revisions to your manuscript. Therefore, I invite you to respond to the comments and revise your manuscript.

Please you submit the revised version of your manuscript within 7 days (i.e. by the 28-Mar-2021). If you do not think you will be able to meet this date please let me know immediately.

When submitting your revised manuscript, you will be able to respond to the comments made by the referees and upload a file "Response to Referees" in the "File Upload" step. You can use this to document any changes you make to the original manuscript. In order to expedite the processing of the revised manuscript, please be as specific as possible in your response to the referees.

Full author guidelines can be found here <https://royalsocietypublishing.org/rsos/replication-studies#AuthorsGuidance>.

Kind regards,
Professor Chris Chambers
Royal Society Open Science
openscience@royalsociety.org

on behalf of Professor Chris Chambers (Registered Reports Editor, Royal Society Open Science)
openscience@royalsociety.org

Associate Editor Comments to Author (Professor Chris Chambers):

Associate Editor: 1

Comments to the Author:

Three expert reviewers have now provided detailed and constructive reviews. The comments are overall very positive, with each reviewer judging the Stage 1 primary criteria to be broadly met. The reviewers also offer a range of helpful suggestions for minor improvements and corrections to the Introduction and framing of prior literature, for clarifying specific aspects of the methodology and analysis plans, and for ensuring that the reporting of the original study is as accurate as possible.

Overall, the manuscript is in good shape and provided the authors are able to respond comprehensively to these points in a revision and response, in-principle acceptance should be forthcoming without requiring further in-depth Stage 1 review.

Reviewers' comments to Author:

Reviewer: 1

Comments to the Author(s)

Background/Summary:

- In a 2013 study, Johnson et al. published a 4-experiment paper which shows that participants are slower to respond to stimuli that were 'refreshed' (a sort-of 'cued reflection') immediately before, compared to stimuli that were not refreshed, but were previously presented. An analogy was made with IOR, an attentional effect in which participants are slower to respond to/fixate on previously-disengaged locations.
- Some participants also completed a memory task, which showed that refreshed-probe trials have improved recall, adding a layer of intrigue to this effect.
- Because this runs contrary to the understanding of how refreshing increases activation, instead of suppresses (re: Johnson), and because this has not been supported by other research (to their knowledge), the current authors look to replicate one of the experiments presented.

Disposition

I believe this is an important and relevant replication to run, especially before using resources to try to understand the time-course/ other properties of this 'IOR-like' effect. I only have a few concerns, which I am confident the authors will be able to effectively address.

Comments and Concerns

- I think you did a great job of outlining your statistics, and additionally how you plan on achieving appropriate power while collecting data.
- The methods are effectively the same, and the authors do a good job justifying any minor differences between the studies.
- I feel as if the authors could have used more citations to support some of the claims being made throughout the introduction. A couple examples:
 - o "One of the main processes being investigated within the field of working memory is "refreshing", i.e., actively thinking back to information that is no longer perceptually present by focusing one's attention upon the internal representation of the information. In terms of attentional processes, refreshing is hence concerned with internal attention (i.e., orienting towards information represented in working memory), while the classical inhibition of return effect is concerned with external attention (i.e., orienting towards information that is perceptually present in the surrounding environment)." I think this may need a few citations.
 - o "Overall, the concept of refreshing in the working memory literature has always been linked to increased activation of working memory representations and, to our knowledge, never to inhibition, except for the studies by Johnson et al. and a follow-up by some of these same authors." I would like to see some of the literature cited earlier in the paper cited again here to support this claim, but I also think that maybe more studies/ information could be included in support of activation
- "Yet, studies showing increased recall performance after refreshing do not necessarily contradict Johnson et al's results..." I don't think that this is worded as accurately as it can be, as there is explicitly no contradiction. Johnson et al. showed improved recall for refreshing, so it is congruent with the literature in this aspect. I think this interesting discrepancy between this IOR-like effect and the LTM improvement for refreshed probes is worth discussing, but it must be done effectively to not confuse/mislead the reader.
- While I generally understand what will constitute a successful replication based on what you have written (you do talk about your 't-test' and such), I think you could more explicitly outline what your hypothesis is, and what exact statistical outcomes will support that hypothesis.

Reviewer: 2

Comments to the Author(s)

To start off, full disclosure: I am Matt Johnson, the first author of the paper being replicated here. I reviewed this manuscript with the help of two of my Ph.D. students, Evan Lintz and Zach Cole. It should be noted that we had some prior familiarity with this line of work and had previously chatted with the authors about it at a conference. Our two research groups have never actually collaborated, so I don't think this is any kind of conflict of interest, but the context may be helpful.

For obvious reasons, we may be a bit biased, but we think generally this line of work is interesting and deserving of replication/extension, so overall we are on board with this effort. We have a few suggestions, questions, and comments, although none of the issues we'll be bringing up are serious enough to be considered deal-breakers for ultimate publication... we are merely trying to do what we can to make the ultimate published paper the best it can be. In no particular order:

1) A general note: There are a number of places where the authors note (fairly) the lack of certain methodological details in the original paper being replicated. We are certainly aware of those omissions... unfortunately, due to the very stringent word count limits of Psychological Science and the numerous requests of the reviewers at the time, we had to be extremely terse throughout the paper to fit everything in, and that required cutting out some of the finer details of the methods during the revision process. Some of these added details can be found in Higgins et al (2020), which the authors cited and which mostly uses the same methods as the original 2013 paper, but even Higgins et al omitted some of them. Thus, we would be happy to provide additional methodological details if the authors feel like including them for comparison would be useful. Of course normally back-channel communication between authors and reviewers is frowned upon, but in this instance since open review is encouraged anyway, I assume it would probably be OK to send an email or two back and forth to request/ deliver any desired details. (If we really want to be on the up-and-up, I guess we could disclose the contents of those messages to the editors, just in case there are any concerns.) Anyway, if the authors prefer to restrict their discussion to the details that have been given in the official peer-reviewed version of the paper, that is fine with us too; we just wanted to make the offer in case it would help improve this paper and/or inform the authors' future work.

(N.b.: Hereafter when we mention page numbers, we'll use the authors' original page numbers, not the page numbers imposed by the manuscript submission system. I.e. when we say page 4, we mean the authors' original page 4, which is labeled "page 6 of 15" in the RSOS header and is actually the 7th page of the PDF file.)

2) We found the framing of some parts of the introduction just a tiny bit odd. For example, in lines 39-44 of the Summary ("Contrary to this study..."), as well as in lines 22-26 on page 4 ("While these two experiments...") and in lines 39-42 on page 5, the authors seem to be setting up the original finding as a bit of a paradox -- why did Johnson et al (2013) seemingly find inhibition when others find facilitation from refreshing? If one accepts the framework our group tends to use in which mental attention is assumed to have many of the same attributes as perceptual attention, it really is not that paradoxical... just like perceptual attention, it seems reasonable that mental attention could involve both short-term and long-term facilitation, but have a brief "refractory period" after the initial facilitation; hence the original comparison to perceptual IOR. (And of course such refractory periods are a fundamental part of how the brain operates, as we can see in both the action potential and in a standard hemodynamic response in fMRI, although we admit those do not necessarily entail that every excitatory mental process must necessarily be followed by a behaviorally observable period of inhibition.) Anyway, this point is not too big of a deal... we just thought that this framing set up the original effect as perhaps a bigger mystery than it really was. (Granted, it surprised us a little bit when we initially found the effect -- our behind-the-scenes story is that we also expected these experiments to produce short-term facilitation and had intended to follow them up with an entirely different line of experiments -- but we came around to the IOR-like viewpoint fairly quickly after that initial surprise. We also do admit that there are some remaining questions around why some study designs show these IOR-like effects and relatively similar designs show facilitation, and we are actually doing some of those experiments ourselves right now.)

3) Again, not particularly necessary, but in the discussion of refreshing producing better memory performance (lines 22-45 on page 4), the authors may want to also mention the work of Marcia Johnson et al (e.g. Johnson et al 2002 -- "Second thoughts versus second looks: An age-related deficit in reflectively refreshing just-active information"), which as far as I'm aware is the earliest example of such findings (at least with designs in which the task/process is called "refreshing" -- similar designs might have existed earlier in cognitive psychology history under different names, but if so I am not aware of them).

4) Once more, not particularly necessary, but we have an in-press paper that might be relevant to the discussion around pages 4-5 of previous facilitative refresh effects. Lintz & Johnson (in press at *Cognition*, Exp 2) also found short-term facilitation for refreshed words in a design fairly close to designs we have previously used that produced IOR-like effects. This is by no means an ego trip and the authors should not feel compelled to cite that paper, but you may find a comparison of that experimental design to other designs useful in trying to work out what are the critical factors that differentiate IOR-like-effect-producing experimental designs from facilitation-producing designs. A preprint is available at <https://psyarxiv.com/z52vf/> if it is not available at *Cognition* in time for you to read it there.

5) Very minor -- typo on line 36 of page 7 -- "The probe could respond to..." should, I think, be "The probe could correspond to..."

6) As noted in point #1, we are happy to provide more detailed methodological information from our original study if the authors want it. However, the good news is that in many respects the authors did more-or-less similar things to our original methods anyway. Some highlights from memory (which as we all know is unreliable -- but I think these details are all correct): We used nouns of 1 or 2 syllables that were not exactly constrained to have all six letters, but which averaged around that same length (I would have to look up the exact number, but am happy to do so if it's wanted). We then also tried to remove highly valenced words and duplicates, although not in exactly the same manner as in this current study, but with similar intentions and (we expect) similar results. For future reference, in case the authors ever want to use it, we balanced our word lists with a tool that we have now improved and made public as the LIBRA toolbox in Matlab (manuscript under review, preprint at <https://psyarxiv.com/64yfw/>). We also used a short training phase which, if I recall correctly, also had six trials. Similarly, we also split the 144 trials into three blocks of 48 with breaks in-between. We also ran the original study on E-Prime, although at the time I believe we were still on version 1. In addition to digital recordings and the Matlab script we used, we actually also used an E-Prime voice key apparatus similar to the authors -- we in fact did do analyses with both the digital recording/Matlab script and with the E-Prime voice key numbers, and found them to largely agree... we just found that the digital recordings gave more precise results (less variance), and those were always intended to be the "primary" measure with the E-Prime as a backup, so with the space constraints of the original article, we only published the digital recording results. (If the authors ever want to do analyses with digital recordings, the script is not yet public but it is fairly user-friendly after a bit of introduction, so we'd be happy to share that too for future work.) Our exclusion criteria were also pretty similar: Although the digital process we used allowed us to keep a few more trials with extraneous noise in them that did not directly involve the task (e.g. noises from inside or outside the testing room from someone moving around, accidentally bumping the microphone, doors closing, etc.), we generally excluded any trials where the noise was during the cue or probe period when speech was supposed to be occurring, including subject-generated noises like sniffs or throat clearing. (However, when the only noise was a small popping sound from the subject's mouth opening, we left the trials in but made sure the detected onset was for the first recognizable phoneme, not the pop of the mouth opening.) The authors are correct in assuming that if either the response during the refresh cue OR the response during the probe was excluded, we excluded the entire trial from analysis. (However, we only reported RTs for the probe period, again due to space constraints; if we had reported RTs for refreshing, I think we would have left in refresh instances where the subject subsequently misspoke on the probe, because the refresh precedes the probe temporally.) I suppose you could say our exclusion criteria for rejecting too-early responses was similar, although since we had the digital recording it was easy to listen back and ascertain the reason for the early trigger; since it is effectively impossible to have a legitimate response onset occur in >150ms, any such instances were either accidental speech sounds (in which case the trial was excluded) or incidental noises like mouth pops or environmental noise (in which case the trial was included, as long as the noise did not appear to interfere with either

the subject's generation of a correct response or our ability to resolve it in the recording). Anyway, this has gone a bit long (although there are plenty more details that have been omitted, but which we'd be happy to provide) – but all in all, I'm not 100% sure it's fair to say that "exclusion criteria were slightly stricter than the ones used in the original paper" (page 10, lines 31-33)... although it is reasonable to see how that conclusion could be reached, given the limited amount of detail we had room to include in the Psych Science paper. It might be fairer to say that the E-Prime voice key vs. the digital recording method simply allow different criteria to be used, because they are fundamentally different in character. With all that said, I think the choices made in the current study using the E-Prime voice key are reasonable and very similar to the criteria we would have used if we didn't have digital recording data ourselves.

7) We mostly found the counterbalancing scheme used here reasonable (see next point though), although it was different from the one we originally used. Briefly, from memory (we can give more details and/or check the veracity of these memories if desired), we effectively generated nine balanced word lists and used a 3x3 counterbalancing scheme, wherein we rotated through conditions (e.g., relative to counterbalance version 1, on counterbalance version 2 all refreshed-probe trials become unrefreshed-probe trials, unrefreshed-probe become novel-probe, and novel-probe become refreshed probe) and also word positions (e.g., relative to counterbalance version 1, on counterbalance version 4 all the initially-top words become initially-bottom words, initially-bottom become novel-position words, and novel-position words become initially-top words). This also leads to some oddities insofar as in each version, 2/9 word lists are not seen (the words ostensibly in the novel-position but on refreshed-probe or unrefreshed-probe trials, where the novel word is "invisible"), but as far as we could tell, it was methodologically sound.

8) For the counterbalancing scheme actually used here – as noted, it seems mostly reasonable to us, but we did find it slightly odd that the authors chose to order the words from highest to lowest frequency instead of going with some kind of randomization/pseudo-randomization scheme that would still control for frequency effects (at least, on average) but without a potentially noticeable increase in word rarity throughout the duration of the experiment. As far as we can tell, this does not cause any actual problems for the interpretation of any results, it's just kind of an unusual choice for this kind of experiment... and if the results are the same as what we remember seeing in preliminary form at a previous conference, it doesn't seem to have caused any issues for successfully replicating the original work. Still, it might be worthy of some additional discussion in the Discussion section of the final paper... and it might be an interesting question in itself to compare effects between block 1 and block 3, to see if the frequency manipulation had any noticeable consequences. We are not requiring that or anything, though; it just might be interesting to check. (Of course, then frequency would also be confounded with the simple passage of time, so it would be hard to say anything definitive... but if there are any hints of something interesting, we could probably help out by doing the same analysis in our original data, to save the authors some effort if they are thinking of pursuing any follow-ups on that question.)

9) The choice of one-sided Bayesian t-tests momentarily caused us to raise an eyebrow, given how one-sided tests are generally frowned upon in the NHST domain these days, even for situations in which they might have been historically considered an appropriate choice. However, after doing a bit of background reading, we have convinced ourselves that one-sided tests don't really deserve the same kind of stigma in the Bayesian domain, so we don't consider it an issue. Still, the authors may want to consider including a bit more justification (just a sentence or so, probably) in order to provide similar assurance to other readers who are not too familiar with Bayesian tests and their interpretations.

I think that's everything we found worthy of comment – again, we generally support this line of work and are mostly providing those notes for the sake of completeness and for the authors'

consideration in preparing the next version of the manuscript; none of them are anything we view as super-critical to the ultimate value of the work or the viability of the final paper. We'll be looking forward to seeing the results, and again, just let us know if you'd like us to provide any other methodological background on the work being replicated.

Best wishes,
Matt Johnson, Zach Cole, and Evan Lintz

Reviewer: 3

Comments to the Author(s)

The paper is generally well written and clear. I have only some minor suggestions to clarify a few points and extend the data-analysis to address some potential confounds.

First, it is important to unambiguously state that the inhibition of return occurs in a paradigm in which the cue is non predictive of the upcoming location of the target. When they are predictive facilitation is generally observed. The way it is written in the text, this feature is not mentioned which makes it seem as if attended locations are inhibited in perception. But this general statement is not true. The effect is observed because of the sudden onset capture caused by the box flashing on the screen.

Second, Figure 1B should present the words in English. After all, it is meant to describe the study of Johnson et al. which was with English speaking participants. The authors have an additional figure for their methods which can illustrate the procedure with French words. The figures could also include timings used for maximal information.

Third, the authors only propose to perform a t-test on reading times. I believe an additional analysis should also be reported that takes condition as a fixed effect and item as a random effect in a mixed effects model. Given the uncertainty regarding reading times for the French words, it is important to make sure that the results are not confounded by some outlier words. Reading times during the refreshing phase could also be analyzed to provide a general baseline of spoken times for the words.

Fourth, the authors should consider whether baseline reading times differ between their study and the ones previously reported in the literature that they are aiming to replicate. This may deserve a brief discussion.

Fifth, I was not sure about this exclusion of the trials with reading times below 150 ms. Wouldn't this affect only previously refreshed trials? because in neutral and unrefreshed trials, there would be no way a delayed speech onset from the refreshing phase could lead to a correct response for analysis. To alleviate these concerns, an analysis including and excluding these trials should be provided, as well as specific information about how many trials were excluded because of this in each design cell.

Author's Response to Decision Letter for (RSOS-210254.R0)

See Appendix A.

Decision letter (RSOS-210254.R1)

Dear Dr Langerock

On behalf of the Editor, I am pleased to inform you that your Manuscript RSOS-210254.R1 entitled "Inhibition-of-return-like effects in working memory? A preregistered replication study of Johnson et al. (2013)." has been accepted in principle for publication in Royal Society Open Science.

You may now progress to Stage 2 and complete the article as approved.

Please note that you must now register your approved protocol on the Open Science Framework (<https://osf.io/rr>), using the 'Submit your approved Registered Report' option and then the 'Registered Report Protocol Preregistration' option. Please use the Registered Report option even though your article is being accepted as a Stage 1 Replication. Further into the registration process, in the Journal Title field enter 'Royal Society Open Science (Replication article type, Results-Blind track)'. Please note that a time-stamped, independent registration of the protocol is mandatory under journal policy, and manuscripts that do not conform to this requirement cannot be considered at Stage 2. The protocol should be registered unchanged from its current approved state. Please include a URL to the protocol in your Stage 2 manuscript, and because you submitted via the Results-Blind track please note in the manuscript that the pre-registration was performed after data analysis (e.g. 'This article received results-blind in-principle acceptance (IPA) at Royal Society Open Science. Following IPA, the accepted Stage 1 version of the manuscript, not including results and discussion, was preregistered on the OSF (URL). This preregistration was performed after data analysis.')

Please note that this registration of the Stage 1 manuscript is required even though you separately preregistered your study protocol. In reporting the registrations in the Stage 2 manuscript, please distinguish them clearly for readers.

Following completion of your manuscript, we invite you to resubmit it for peer review as a Stage 2 Replication. Please note that your manuscript can still be rejected for publication at Stage 2 if the Editors consider any of the following conditions to be met:

- The Introduction and methods deviated from the approved Stage 1 submission (required).
- The authors' conclusions were not considered justified given the data.

We encourage you to read the complete guidelines for authors concerning Stage 2 submissions at <https://royalsocietypublishing.org/rsos/replication-studies#AuthorsGuidance>. Please especially note the requirements for data sharing and that withdrawing your manuscript will result in publication of a Withdrawn Registration.

Once again, thank you for submitting your manuscript to Royal Society Open Science and I look forward to receiving your Stage 2 submission. If you have any questions at all, please do not hesitate to get in touch. We look forward to hearing from you shortly with the anticipated submission date for your stage two manuscript.

Kind regards,
Professor Chris Chambers

on behalf of Professor Chris Chambers (Registered Reports Editor, Royal Society Open Science)
openscience@royalsociety.org

Author's Response to Decision Letter for (RSOS-210254.R1)

See Appendix B.

RSOS-210254.R2 (Revision)

Review form: Reviewer 1 (Colin McCormick)

Do you have any ethical concerns with this paper?

No

Have you any concerns about statistical analyses in this paper?

No

Recommendation?

Accept with minor revision

Comments to the Author(s)

File with comments is attached (see Appendix C).

Review form: Reviewer 2 (Matthew Johnson)

Do you have any ethical concerns with this paper?

No

Have you any concerns about statistical analyses in this paper?

Yes

Recommendation?

Major revision

Comments to the Author(s)

Hello again. It is one of your reviewers from last time, Matt Johnson, still with the help of my Ph.D. students Evan Lintz and Zach Cole. I think we were clear before that we are enthusiastic about this line of work generally, and we offered most of what we have to say on the previous round of reviews before results were added in, so we'll just jump right in and try to be brief.

Major comments:

Only one major one here. This may not end up being an issue at all, but we wanted to make sure. If we are understanding everything correctly, there seems to be a curiously large jump in the Bayes factor for the primary hypothesis when going from $N=28$ (2.54) to $N=33$ (28). Of course almost anything is statistically possible in the unpredictable world of human subjects data, but it seemed intuitively unlikely so we dug in a bit further. We would have tried working with the actual data, but it appears it is not yet available at the OSF link included in the paper? (This is perfectly fine if the authors want to wait to make it available until after acceptance... we just wanted to note that we tried that option first.) So next we ran some simulations to generate a few datasets that had a BF of around 2.5 at $N=28$, and then tried to assess what the likelihood would be of observing a $BF \geq 28$ after adding just five more subjects drawn randomly from the same underlying distribution. Naturally a number of assumptions go into such simulations (e.g. that values are normally distributed and so on), so it is hard to know how well they approximate reality... but in any case, it seemed like adding five subjects and seeing the BF jump from ~ 2.5 to ~ 28 was exceedingly rare (on the order of 1 out of every 10,000 simulations or so). Anyway, this is a long way of saying... is it possible that the BF of 28 at $N=33$ could be a typo or some other kind of error? If you are confident in those values, then that is great and we're sorry to waste your time... but the difference was striking enough that hopefully you can understand our surprise.

Minor comments

(Note that there are several ways to number pages. The authors appear to have originally numbered odd-numbered pages. It looks like the submission system added its own numbering system to the proof. And the page within the PDF file is also different. For example, in our proof, the page the authors originally numbered as 7 is printed in the RSOS header as "Page 9 of 18" and it is the 10th page in the PDF file. We will use the page numbers in the RSOS header in our notes below, e.g. "Page 9 of 18")

page 4, line 55: Typo... the name should be Kiyonaga

page 5, line 11: It could be confusing to say words were presented "vertically," which sort of sounds like the individual letters in the words were printed in a vertical line, rather than the intended meaning (that both words were printed normally but one was vertically above the other). So you may wish to rephrase for clarity.

page 7, line 54: The usage of the word "only" is a bit out of place here, since $2000 > 1600$. We recommend just removing the word "only" on this line.

page 12, "Additional exploratory analyses" section: This section still is written as if it is in a manuscript where the results are not yet known, so it sounds a bit odd to say things like "In case our study does not replicate... the effect..." You could potentially take out this section entirely, since it ultimately appears to have been irrelevant... or if you want to leave it in, it might be good to rephrase it a bit to clarify that those steps WOULD have been taken if the replication did not occur. (This might just be a simple issue of verb tenses/moods.)

References -- the Lintz & Johnson paper is now published, not merely forthcoming, so you can update this reference. (Lintz EN, Johnson MR. 2021. Refreshing and removing items in working memory: Different approaches to equivalent processes? *Cognition* 211: article 104655.)

We hope these comments were helpful, and we remain enthusiastic about this line of work!

Best wishes,
Matt, Evan, and Zach

Review form: Reviewer 3

Do you have any ethical concerns with this paper?

No

Have you any concerns about statistical analyses in this paper?

No

Recommendation?

Accept with minor revision

Comments to the Author(s)

In general, the paper is well-written and the analysis and results were clear cut.

The only pending issue in view was that the data and materials were not really accessible as required by the journal. There was no data available in the OSF page indicated. It was completely blank. I was confused if this was a feature required by the journal - but then, they ask us to evaluate whether the data-posting was clear, which it is not possible as it is.

Decision letter (RSOS-210254.R2)

Dear Dr Langerock,

The editors assigned to your paper ("Inhibition-of-return-like effects in working memory? A preregistered replication study of Johnson et al. (2013).") have now received comments from reviewers. We would like you to revise your paper in accordance with the referee and Subject Editor suggestions which can be found below (not including confidential reports to the Editor).

Please submit a copy of your revised paper within three weeks (i.e. by the @@author due date will be populated when the email is sent@@).

To revise your manuscript, log into <http://mc.manuscriptcentral.com/rsos> and enter your Author Centre, where you will find your manuscript title listed under "Manuscripts with Decisions." Under "Actions," click on "Create a Revision." Your manuscript number has been appended to denote a revision. Revise your manuscript and upload a new version through your Author Centre. Please see <https://royalsocietypublishing.org/rsos/replication-studies#AuthorsGuidance> for guidance.

In addition to addressing all of the reviewers' and editor's comments please also ensure that your revised manuscript contains the following sections as appropriate before the reference list (a non-exhaustive example is included as an attachment):

- Data accessibility

<http://datadryad.org/submit?journalID=RSOS&manu=RSOS-210254.R2>

- Competing interests

- Authors' contributions

- Acknowledgements

- Funding statement

Kind regards,
 Professor Chris Chambers
 Royal Society Open Science
 openscience@royalsociety.org

Associate Editor's comments (Professor Chris Chambers):

The three original reviewers from Stage 1 returned to assess the Stage 2 manuscript, and their evaluations are helpful and broadly positive. The reviewers note a range of areas that would benefit from clarification or elaboration, which I won't summarise here, but the main issue to address is the point raised by Reviewer 2 concerning the accuracy of the results. All of the reviewers note that the data are not (yet) available on the OSF and Reviewer 2 questions whether there may be a reporting error in the Bayesian analyses. Please attend carefully to this point and check that all reporting is accurate, and I would ask also that full open data and code are made available with the next revision so that, if necessary, we can engage the reviewers for any rechecking.

Reviewers' Comments to Author:

Reviewer: 1

Comments to the Author(s)

File with comments is attached.

Reviewer: 2

Comments to the Author(s)

Hello again. It is one of your reviewers from last time, Matt Johnson, still with the help of my Ph.D. students Evan Lintz and Zach Cole. I think we were clear before that we are enthusiastic about this line of work generally, and we offered most of what we have to say on the previous round of reviews before results were added in, so we'll just jump right in and try to be brief.

Major comments:

Only one major one here. This may not end up being an issue at all, but we wanted to make sure. If we are understanding everything correctly, there seems to be a curiously large jump in the Bayes factor for the primary hypothesis when going from $N=28$ (2.54) to $N=33$ (28). Of course almost anything is statistically possible in the unpredictable world of human subjects data, but it seemed intuitively unlikely so we dug in a bit further. We would have tried working with the actual data, but it appears it is not yet available at the OSF link included in the paper? (This is perfectly fine if the authors want to wait to make it available until after acceptance... we just wanted to note that we tried that option first.) So next we ran some simulations to generate a few datasets that had a BF of around 2.5 at $N=28$, and then tried to assess what the likelihood would be of observing a $BF \geq 28$ after adding just five more subjects drawn randomly from the same underlying distribution. Naturally a number of assumptions go into such simulations (e.g. that values are normally distributed and so on), so it is hard to know how well they approximate reality... but in any case, it seemed like adding five subjects and seeing the BF jump from ~ 2.5 to ~ 28 was exceedingly rare (on the order of 1 out of every 10,000 simulations or so). Anyway, this is a long way of saying... is it possible that the BF of 28 at $N=33$ could be a typo or some other kind of error? If you are confident in those values, then that is great and we're sorry to waste your time... but the difference was striking enough that hopefully you can understand our surprise.

Minor comments

(Note that there are several ways to number pages. The authors appear to have originally numbered odd-numbered pages. It looks like the submission system added its own numbering system to the proof. And the page within the PDF file is also different. For example, in our proof, the page the authors originally numbered as 7 is printed in the RSOS header as "Page 9 of 18" and it is the 10th page in the PDF file. We will use the page numbers in the RSOS header in our notes below, e.g. "Page 9 of 18")

page 4, line 55: Typo... the name should be Kiyonaga

page 5, line 11: It could be confusing to say words were presented "vertically," which sort of sounds like the individual letters in the words were printed in a vertical line, rather than the intended meaning (that both words were printed normally but one was vertically above the other). So you may wish to rephrase for clarity.

page 7, line 54: The usage of the word "only" is a bit out of place here, since $2000 > 1600$. We recommend just removing the word "only" on this line.

page 12, "Additional exploratory analyses" section: This section still is written as if it is in a manuscript where the results are not yet known, so it sounds a bit odd to say things like "In case our study does not replicate... the effect..." You could potentially take out this section entirely, since it ultimately appears to have been irrelevant... or if you want to leave it in, it might be good to rephrase it a bit to clarify that those steps WOULD have been taken if the replication did not occur. (This might just be a simple issue of verb tenses/moods.)

References -- the Lintz & Johnson paper is now published, not merely forthcoming, so you can update this reference. (Lintz EN, Johnson MR. 2021. Refreshing and removing items in working memory: Different approaches to equivalent processes? *Cognition* 211: article 104655.)

We hope these comments were helpful, and we remain enthusiastic about this line of work!

Best wishes,
Matt, Evan, and Zach

Reviewer: 3
Comments to the Author(s)

In general, the paper is well-written and the analysis and results were clear cut. The only pending issue in view was that the data and materials were not really accessible as required by the journal. There was no data available in the OSF page indicated. It was completely blank. I was confused if this was a feature required by the journal - but then, they ask us to evaluate whether the data-posting was clear, which it is not possible as it is.

Author's Response to Decision Letter for (RSOS-210254.R2)

See Appendix D.

Decision letter (RSOS-210254.R3)

Dear Dr Langerock:

It is a pleasure to accept your manuscript entitled "Inhibition-of-return-like effects in working memory? A preregistered replication study of Johnson et al. (2013)." in its current form for publication in Royal Society Open Science.

You can expect to receive a proof of your article in the near future. Please contact the editorial office (openscience@royalsociety.org) and the production office (openscience_proofs@royalsociety.org) to let us know if you are likely to be away from e-mail contact – if you are going to be away, please nominate a co-author (if available) to manage the proofing process, and ensure they are copied into your email to the journal.

Appendix A

Dear Professor Chambers,

The manuscript we are submitting today is a revision of Manuscript RSOS-210254 as invited by your decision letter. To remind you, our study aims at replicating inhibition of return in working memory. We thank you and the reviewers for the positive feedback as well as for the constructive comments and suggestions to improve the quality of our manuscript. We appreciate the opportunity to incorporate these. We have carefully addressed each comment/suggestion, and made changes in the manuscript where necessary. Below, we describe how we addressed each of the points raised by the reviewers (our responses are in blue, in the manuscript the modifications are highlighted in yellow).

We hope that this Stage 1 registered replication study is now satisfying the requirements for in-principle-acceptance in Royal Society Open Science.

Kind Regards,

Naomi Langerock, Giuliana Sposito, Caro Hautekiet & Evie Vergauwe

Response to reviewers

Reviewer 1

1. I feel as if the authors could have used more citations to support some of the claims being made throughout the introduction. A couple examples:

1.1 “One of the main processes being investigated within the field of working memory is “refreshing”, i.e., actively thinking back to information that is no longer perceptually present by focusing one’s attention upon the internal representation of the information. In terms of attentional processes, refreshing is hence concerned with internal attention (i.e., orienting towards information represented in working memory), while the classical inhibition of return effect is concerned with external attention (i.e., orienting towards information that is perceptually present in the surrounding environment).” I think this may need a few citations.

We agree and have now added the following citations in this sentence, on p. 3:

8. Johnson, MK. MEM: Mechanisms of recollection. *J Cogn Neurosc*. 1992 ; **4(3)** : 268-280. doi: 10.1162/jocn.1992.4.3.268
9. Camos V, Johnson M, Loaiza V, Portrat S, Souza A, Vergauwe E.. What is attentional refreshing in working memory? *Ann NY Acad Sci*. 2018; **1424(1)**: 19–32. doi: 10.1111/nyas.13616
10. Chun, M. Visual working memory as visual attention sustained internally over time. *Neuropsychologia*. 2011; **49(6)**: 1407-1409. doi: 10.1016/j.neuropsychologia.2011.01.029
11. Kiyonaga A, Egner T. Working memory as internal attention: Toward an integrative account of internal and external selection processes. *Psychon Bull Rev*. 2013; **20(2)**: 228–42. doi: 10.3758/s13423-012-0359-y
12. Verschooren S, Schindler S. De Raedt R. Pourtois G. Switching attention from internal to external information processing: A review of the literature and empirical support of the resource sharing account. *Psychonom Bull Rev*. 2019; **26(2)**: 468-490.

This sentence reads now:

“One of the main processes being investigated within the field of working memory is “refreshing”, i.e., actively thinking back to information that is no longer perceptually present by focusing one’s attention upon the internal representation of the information^{8,9}. In terms of attentional processes, refreshing is hence concerned with internal attention (i.e., orienting towards information represented in working

memory), while the classical inhibition of return effect is concerned with external attention (i.e., orienting towards information that is perceptually present in the surrounding environment; see e.g., Chun¹⁰, Kiyonagy et al.¹¹ or Verschooren et al.¹² for similar conceptions of internal versus external attention).”

1.2 “Overall, the concept of refreshing in the working memory literature has always been linked to increased activation of working memory representations and, to our knowledge, never to inhibition, except for the studies by Johnson et al. and a follow-up by some of these same authors.” I would like to see some of the literature cited earlier in the paper cited again here to support this claim, but I also think that maybe more studies/information could be included in support of activation.

We agree and have now added the following citations in this sentence, on p. 6 .

8. Johnson, MK. MEM: Mechanisms of recollection. *J Cogn Neurosc*, 1992 ; **4(3)** : 268-280. doi: 10.1162/jocn.1992.4.3.268

9. Camos V, Johnson M, Loaiza V, Portrat S, Souza A, Vergauwe E. What is attentional refreshing in working memory? *Ann NY Acad Sci*. 2018; **1424(1)**:19–32. doi: 10.1111/nyas.13616

24. Cowan N. An embedded processes model of working memory. In: Miyake A, Shah P, editors. *Models of Working Memory: Mechanisms of active maintenance and executive control*. Cambridge: Cambridge University Press; 1999. p. 62-101.

25. Raye CL, Johnson MK, Mitchell KJ, Greene EJ, Johnson MR. Refreshing: A minimal executive function. *Cortex*, 2007; **43(1)**: 135-145.

This sentence reads now: “Overall, the concept of refreshing in the working memory literature has always been linked to increased activation of working memory representations^{8,9,13-15,24,25}”

2. “Yet, studies showing increased recall performance after refreshing do not necessarily contradict Johnson et al’s results...” I don’t think that this is worded as accurately as it can be, as there is explicitly no contradiction. Johnson et al. showed improved recall for refreshing, so it is congruent with the literature in this aspect. I think this interesting discrepancy between this IOR-like effect and the LTM improvement for refreshed probes is worth discussing, but it must be done effectively to not confuse/mislead the reader.

We agree with the reviewer and have adapted the wording in the following way, on p. 4 :

“ While Johnson et al.’s¹ studies show inhibitory effects immediately after refreshing, they also show increased memory performance in a long term memory test. In their experiment in the verbal domain (words), a surprise memory test was administered after about 20 minutes of performing another task (Experiment 1A). On this long-term memory test, better memory performance was observed for the refreshed items (independently of whether they had been the probed item afterwards or not), showing hence that refreshing boosts long-term memory performance. This is in line with previous studies that have shown that refreshing of memory items may boost memory performance not only for working memory but also for long-term memory¹⁶⁻¹⁸. The studies by Johnson et al.¹ showed hence a combination of inhibitory processes right after refreshing and increased recall performance in the long term.”

3. While I generally understand what will constitute a successful replication based on what you have written (you do talk about your ‘t-test’ and such), I think you could more explicitly outline what your hypothesis is, and what exact statistical outcomes will support that hypothesis.

We agree and have adapted the last paragraph of the Introduction (p. 6) to include this information explicitly. This reads now:

“ The present study reports our replication of Experiment 1B reported by Johnson et al.¹. In this experiment we test the hypothesis that words that have been refreshed just before are slower to access than words that have not been refreshed just before. The replication of this study was preregistered on

aspredicted.org (<https://aspredicted.org/s82zc.pdf>) and the results and materials for this replication can be found on OSF (<https://osf.io/3k6rv/>). The method and analysis section of Experiment 1B reported by Johnson et al.¹ were followed as closely as possible, although we tested the effect in a French-speaking population instead of an English-speaking population and we used Bayesian sequential hypothesis testing instead of null hypothesis significance testing to examine if an inhibition-of-return-like effect is observed after refreshing in working memory. Bayes Factors (BF) represent evidence in the data in favor or against a specific hypothesis, and evidence is considered to be moderate as from a BF of three on, in favor or against the hypothesis²⁶. In the present experiment, the main hypothesis is tested by a one-sided Bayesian t-test because we have specific predictions about the direction of the hypothesized effect, and we aimed for a BF of 10 in favor of (or against) the hypothesis, which is considered strong evidence. “

Furthermore, on p. 11 in the paragraph on the planned preregistered analysis, we now also state:” This latter t-test concerned our main hypothesis.”

Reviewer 2

1. A general note: There are a number of places where the authors note (fairly) the lack of certain methodological details in the original paper being replicated. We are certainly aware of those omissions... unfortunately, due to the very stringent word count limits of Psychological Science and the numerous requests of the reviewers at the time, we had to be extremely terse throughout the paper to fit everything in, and that required cutting out some of the finer details of the methods during the revision process. Some of these added details can be found in Higgins et al (2020), which the authors cited and which mostly uses the same methods as the original 2013 paper, but even Higgins et al omitted some of them. Thus, we would be happy to provide additional methodological details if the authors feel like including them for comparison would be useful. Of course normally back-channel communication between authors and reviewers is frowned upon, but in this instance since open review is encouraged anyway, I assume it would probably be OK to send an email or two back and forth to request/deliver any desired details. (If we really want to be on the up-and-up, I guess we could disclose the contents of those messages to the editors, just in case there are any concerns.) Anyway, if the authors prefer to restrict their discussion to the details that have been given in the official peer-reviewed version of the paper, that is fine with us too; we just wanted to make the offer in case it would help improve this paper and/or inform the authors' future work.

We thank the reviewers for proposing to share all the necessary methodological information. Reading further through their comments/suggestions, it appears that none of our methodological decisions deviates much from what has been done by the reviewers in their previous studies. At this point, we think it is reasonable to stick to these small deviations, based on what the original, peer-reviewed article disclosed. If it turns out that we could not replicate the IOR-like effect, these small deviations would become of higher importance and we would take the opportunity to further discuss these deviations with the reviewers and potentially include a discussion of these in the discussion of the Phase 2 manuscript.

2. We found the framing of some parts of the introduction just a tiny bit odd. For example, in lines 39-44 of the Summary ("Contrary to this study..."), as well as in lines 22-26 on page 4 ("While these two experiments...") and in lines 39-42 on page 5, the authors seem to be setting up the original finding as a bit of a paradox -- why did Johnson et al (2013) seemingly find inhibition when others find facilitation

from refreshing? If one accepts the framework our group tends to use in which mental attention is assumed to have many of the same attributes as perceptual attention, it really is not that paradoxical... just like perceptual attention, it seems reasonable that mental attention could involve both short-term and long-term facilitation, but have a brief "refractory period" after the initial facilitation; hence the original comparison to perceptual IOR. (And of course such refractory periods are a fundamental part of how the brain operates, as we can see in both the action potential and in a standard hemodynamic response in fMRI, although we admit those do not necessarily entail that every excitatory mental process must necessarily be followed by a behaviorally observable period of inhibition.) Anyway, this point is not too big of a deal... we just thought that this framing set up the original effect as perhaps a bigger mystery than it really was. (Granted, it surprised us a little bit when we initially found the effect -- our behind-the-scenes story is that we also expected these experiments to produce short-term facilitation and had intended to follow them up with an entirely different line of experiments -- but we came around to the IOR-like viewpoint fairly quickly after that initial surprise. We also do admit that there are some remaining questions around why some study designs show these IOR-like effects and relatively similar designs show facilitation, and we are actually doing some of those experiments ourselves right now.)

We understand the reviewers' point. However, we would like to point out that, when going through the working memory literature, we have not come across studies showing refreshing to result in inhibitory processes (except for the study we are replicating here, and the related studies from Johnson and colleagues). In contrast, most studies link refreshing to an increase of activation, and describe the process as one that reactivates, or boosts working memory representations. In that sense, we think that highlighting this discrepancy between the refreshing literature and the line of research showing inhibitory effects (Johnson 2013 and Higgins 2020) is both appropriate and reasonable. To us, this discrepancy is the main reason why studies showing inhibitory effects are of particular interest, and why we aimed at replicating the effect in our lab. In the same way as the reviewers state: "it surprised us a little bit when we initially found the effect", we were surprised when reading the original article showing inhibitory effects of refreshing. Moreover, we remain surprised by these results, especially because 1) some of our own studies have shown the expected facilitative effects of refreshing on item accessibility (e.g., Vergauwe & Langerock, 2017 in *Journal of Memory and Language*: "Attentional refreshing of information in working memory: Increased accessibility of just-refreshed representations.") and 2) follow-up studies from the reviewers' lab observed facilitatory effects (we refer to the article in press for which the authors had the kindness to provide the link to the preprint version (<https://psyarxiv.com/z52vf/>)). For these reasons, in the light of the currently available findings, we think it is appropriate and reasonable to present the current state of affairs as contrasting results or even as a paradox.

3. Again, not particularly necessary, but in the discussion of refreshing producing better memory performance (lines 22-45 on page 4), the authors may want to also mention the work of Marcia Johnson et al (e.g. Johnson et al 2002 -- "Second thoughts versus second looks: An age-related deficit in reflectively refreshing just-active information"), which as far as I'm aware is the earliest example of such findings (at least with designs in which the task/process is called "refreshing" -- similar designs might have existed earlier in cognitive psychology history under different names, but if so I am not aware of them).

In fact, this reference occurs in the paragraph following the one the reviewers mention. We thought it fitted better with the better memory performance observed in long term memory as a result of refreshing. The memory test in question is presented as Phase 2 in Johnson et al. (2002) and is presented 5 minutes after the end of Phase 1. This is a rather typical set-up when researchers aim at testing the

long-term effects of refreshing, as opposed to the effects on immediate memory. The paragraph the reviewers are referring to considers the effects of refreshing on immediate/working memory while the following paragraph considers the effects of refreshing on long term memory. Therefore, the Johnson et al. (2002) fits better in the second paragraph, the one discussing effects of refreshing on long term memory.

We have added the words long term memory in the first line on the paragraph regarding the long term memory effects, on p. 4, to make this distinction between the two paragraphs more apparent: *“While Johnson et al.’s¹ studies show inhibitory effects immediately after refreshing, they do show increased memory performance on a long term memory test”*

4. Once more, not particularly necessary, but we have an in-press paper that might be relevant to the discussion around pages 4-5 of previous facilitative refresh effects. Lintz & Johnson (in press at Cognition, Exp 2) also found short-term facilitation for refreshed words in a design fairly close to designs we have previously used that produced IOR-like effects. This is by no means an ego trip and the authors should not feel compelled to cite that paper, but you may find a comparison of that experimental design to other designs useful in trying to work out what are the critical factors that differentiate IOR-like-effect-producing experimental designs from facilitation-producing designs. A preprint is available at [/psyarxiv.com/z52vf/](https://psyarxiv.com/z52vf/) if it is not available at Cognition in time for you to read it there.

We thank the reviewers for sharing this very valuable information. We have added this information in the introduction on p. 5 and will most certainly include this information in the discussion. In the introduction, this reads now:

“In accordance with the results of this study by Vergauwe et al, a recent study²⁰ using a paradigm quite similar to the IOR paradigm in working memory also observed facilitatory effects for the refreshed items. So despite the similarities in the paradigm used, no inhibition-of-return-like effects were observed in this study.”

5. Very minor -- typo on line 36 of page 7 -- "The probe could respond to..." should, I think, be "The probe could correspond to..."

Thanks for noticing, we have corrected this typo.

6. As noted in point #1, we are happy to provide more detailed methodological information from our original study if the authors want it. However, the good news is that in many respects the authors did more-or-less similar things to our original methods anyway. Some highlights from memory (which as we all know is unreliable -- but I think these details are all correct): We used nouns of 1 or 2 syllables that were not exactly constrained to have all six letters, but which averaged around that same length (I would have to look up the exact number, but am happy to do so if it's wanted). We then also tried to remove highly valenced words and duplicates, although not in exactly the same manner as in this current study, but with similar intentions and (we expect) similar results. For future reference, in case the authors ever want to use it, we balanced our word lists with a tool that we have now improved and made public as the LIBRA toolbox in Matlab (manuscript under review, preprint at <https://psyarxiv.com/64yfw/>). We also used a short training phase which, if I recall correctly, also had six trials. Similarly, we also split the 144 trials into three blocks of 48 with breaks in-between. We also ran the original study on E-Prime, although at the time I believe we were still on version 1. In addition to digital recordings and the Matlab script we used, we actually also used an E-Prime voice key apparatus similar to the authors -- we in fact did do analyses with both the digital recording/Matlab script and with the E-Prime voice key numbers, and found them to largely agree... we just found that the digital recordings gave more precise results

(less variance), and those were always intended to be the "primary" measure with the E-Prime as a backup, so with the space constraints of the original article, we only published the digital recording results. (If the authors ever want to do analyses with digital recordings, the script is not yet public but it is fairly user-friendly after a bit of introduction, so we'd be happy to share that too for future work.) Our exclusion criteria were also pretty similar: Although the digital process we used allowed us to keep a few more trials with extraneous noise in them that did not directly involve the task (e.g. noises from inside or outside the testing room from someone moving around, accidentally bumping the microphone, doors closing, etc.), we generally excluded any trials where the noise was during the cue or probe period when speech was supposed to be occurring, including subject-generated noises like sniffs or throat clearing. (However, when the only noise was a small popping sound from the subject's mouth opening, we left the trials in but made sure the detected onset was for the first recognizable phoneme, not the pop of the mouth opening.) The authors are correct in assuming that if either the response during the refresh cue OR the response during the probe was excluded, we excluded the entire trial from analysis. (However, we only reported RTs for the probe period, again due to space constraints; if we had reported RTs for refreshing, I think we would have left in refresh instances where the subject subsequently misspoke on the probe, because the refresh precedes the probe temporally.) I suppose you could say our exclusion criteria for rejecting too-early responses was similar, although since we had the digital recording it was easy to listen back and ascertain the reason for the early trigger; since it is effectively impossible to have a legitimate response onset occur in >150ms, any such instances were either accidental speech sounds (in which case the trial was excluded) or incidental noises like mouth pops or environmental noise (in which case the trial was included, as long as the noise did not appear to interfere with either the subject's generation of a correct response or our ability to resolve it in the recording). Anyway, this has gone a bit long (although there are plenty more details that have been omitted, but which we'd be happy to provide) -- but all in all, I'm not 100% sure it's fair to say that "exclusion criteria were slightly stricter than the ones used in the original paper" (page 10, lines 31-33)... although it is reasonable to see how that conclusion could be reached, given the limited amount of detail we had room to include in the Psych Science paper. It might be fairer to say that the E-Prime voice key vs. the digital recording method simply allow different criteria to be used, because they are fundamentally different in character. With all that said, I think the choices made in the current study using the E-Prime voice key are reasonable and very similar to the criteria we would have used if we didn't have digital recording data ourselves.

We thank the reviewers for sharing this information. On the general level, it appears that our methods do not differ much from the original study, and we are glad to see we made similar decisions on several aspects. We have used the E-prime voice key (and not the matlab digital recording) because of ethical concerns. The local ethical committee tends to allow the use of recordings when these are strictly necessary for the study objectives. We did not consider recordings to be necessary for the study as the E-prime voice-key provides us with the needed response time (= latency to read), and this is the only measure taken into account for the analysis in the original manuscript. We are glad to hear that, overall, both methods give rise to similar results and that the use of response times as measured with the E-prime voice key can be deemed as valid. Due to the additional criteria of the "at least 75% accuracy per participant" and the all-or-nothing elimination of trials (without the possibility to relisten to the voice recording and, on some occasions, decide to include the trial anyhow), we think we can reasonably argue that our exclusion criteria were slightly stricter. In line with the reviewers' suggestions, we have nuanced the following phrase on p. 10 :
"As such, our trial-by-trial exclusion criteria may have been slightly stricter", as this paragraph only concerns the methodological difference between the matlab digital recordings and the e-prime voice-key.

7. We mostly found the counterbalancing scheme used here reasonable (see next point though), although it was different from the one we originally used. Briefly, from memory (we can give more details and/or check the veracity of these memories if desired), we effectively generated nine balanced word lists and used a 3x3 counterbalancing scheme, wherein we rotated through conditions (e.g., relative to counterbalance version 1, on counterbalance version 2 all refreshed-probe trials become unrefreshed-probe trials, unrefreshed-probe become novel-probe, and novel-probe become refreshed probe) and also word positions (e.g., relative to counterbalance version 1, on counterbalance version 4 all the initially-top words become initially-bottom words, initially-bottom become novel-position words, and novel-position words become initially-top words). This also leads to some oddities insofar as in each version, 2/9 word lists are not seen (the words ostensibly in the novel-position but on refreshed-probe or unrefreshed-probe trials, where the novel word is "invisible"), but as far as we could tell, it was methodologically sound.

Thank you for sharing this information. We agree that both counterbalancing schemes are reasonable and acceptable.

8. For the counterbalancing scheme actually used here -- as noted, it seems mostly reasonable to us, but we did find it slightly odd that the authors chose to order the words from highest to lowest frequency instead of going with some kind of randomization/pseudo-randomization scheme that would still control for frequency effects (at least, on average) but without a potentially noticeable increase in word rarity throughout the duration of the experiment. As far as we can tell, this does not cause any actual problems for the interpretation of any results, it's just kind of an unusual choice for this kind of experiment... and if the results are the same as what we remember seeing in preliminary form at a previous conference, it doesn't seem to have caused any issues for successfully replicating the original work. Still, it might be worthy of some additional discussion in the Discussion section of the final paper... and it might be an interesting question in itself to compare effects between block 1 and block 3, to see if the frequency manipulation had any noticeable consequences. We are not requiring that or anything, though; it just might be interesting to check. (Of course, then frequency would also be confounded with the simple passage of time, so it would be hard to say anything definitive... but if there are any hints of something interesting, we could probably help out by doing the same analysis in our original data, to save the authors some effort if they are thinking of pursuing any follow-ups on that question.)

Thank you for pointing this out. The reason for using this word-frequency-based fixed order was twofold. First, we did not want to counterbalance for a confounding factor that can perfectly be controlled. It seems reasonable that response times to high frequent words are shorter than low frequent words (see for example Gerhand, S., & Barry, C. ; 1998). By ordering both the general word list (used for the word pairs, and hence as well the refreshed and unrefreshed probe) as well as the novel probes, we exclude the possibility that some probe words are read faster than others, based on their frequency. In our experiment, the refreshed word, the unrefreshed word, and the novel words are very similar in terms of word frequency on every trial. Other options could have been used to avoid having word-frequency as a confounding factor, yet our method seemed like a reasonable, elegant, and effortless solution. As pointed out by the reviewers, there is no reason to assume that this decision would have any implications for the interpretation of the results. Second, we needed a fixed order for our trial-by-trial word presentation as the experimenter had to manually and online (as opposed to offline, using the recordings) verify for every trial whether the participant refreshed the correct word and read aloud the correct probe word. Having a fixed list facilitates this task drastically and is less error-prone than other options that we had considered. We have now added some explanation in the method section on p. 9:

“Additionally, the use of fixed lists allowed the experimenter to have a printed list of the to be refreshed words and the to be read aloud words, which drastically facilitates scoring the accuracy of these words during the experiment.”. Finally, we think the proposed analysis in terms of the different blocks may be confounded with different kinds of other task effects (like learning effects or fatigue effects) so we choose not to integrate these in the manuscript.

9. The choice of one-sided Bayesian t-tests momentarily caused us to raise an eyebrow, given how one-sided tests are generally frowned upon in the NHST domain these days, even for situations in which they might have been historically considered an appropriate choice. However, after doing a bit of background reading, we have convinced ourselves that one-sided tests don't really deserve the same kind of stigma in the Bayesian domain, so we don't consider it an issue. Still, the authors may want to consider including a bit more justification (just a sentence or so, probably) in order to provide similar assurance to other readers who are not too familiar with Bayesian tests and their interpretations.

The use of Bayesian inference implies that the evidence for and against the null hypothesis, as well as for or against the alternative hypothesis can be assessed. To assess the evidence for an alternative hypothesis that has a specific direction, a one-sided test is appropriate. After all, we do not aim to test whether there is *any* difference in the response times between refreshed and unrefreshed words; our tested hypothesis has clearly direction: the inhibition hypothesis predicts longer response times for refreshed words than for unrefreshed words. Accordingly, and as preregistered, a one-sided t-test was used to assess the evidence in the data for inhibition. On p. 7, right before the start of the method section, we have added the following:

“and we used Bayesian sequential hypothesis testing instead of null hypothesis significance testing to examine if an inhibition-of-return-like effect is observed after refreshing in working memory. Bayes Factors (BF) represent evidence in the data in favor or against a specific hypothesis, and evidence is considered to be moderate as from a BF of three on, in favor or against the hypothesis²⁶. In the present experiment, the main hypothesis is tested by a one-sided Bayesian t-test because we have specific predictions about the direction of the hypothesized effect and we aimed for a BF of 10 in favor of (or against) the hypothesis, which is considered strong evidence.”

Reviewer 3

1. It is important to unambiguously state that the inhibition of return occurs in a paradigm in which the cue is non predictive of the upcoming location of the target. When they are predictive facilitation is generally observed. The way it is written in the text, this feature is not mentioned which makes it seem as attended locations are inhibited in perception. But this general statement is not true. The effect is observed because of the sudden onset capture caused by the box flashing on the screen.

This is indeed an important point. This information about the non-validity of the cue is implicitly present in experimental set-up of the original IOR in perception, but we have now made this information more explicit by adding the following on p. 2 of the manuscript:

“even though the cued box has no predictive value regarding the location of the upcoming target object.”

2. Figure 1B should present the words in English. After all, it is meant to describe the study of Johnson et al. which was with English speaking participants. The authors have an additional figure for their methods which can illustrate the procedure with French words. The figures could also include timings used for maximal information.

We agree that Figure 1B should present the English words and have modified this figure accordingly. The text on p. 3 has also been adapted to include the English and not the French words. It is worth noting that we deliberately choose not to include the timings in Figure 1, and only in Figure 2. The goal of Figure 1 is to show how the two IOR paradigms map nicely onto each other. The timings between the two IOR designs are different, as the fast design timings in the IOR paradigm in perception would not allow for a correct execution of the task in the working memory IOR paradigm, but other than that, the paradigms are highly similar. We think that adding these temporal differences in the Figure, this would draw the attention away from these similarities which we wanted to demonstrate with this figure. We have nevertheless added the information regarding the different temporal parameters in the current version of the manuscript, in the text on p. 3:

“The temporal parameters of the original paradigm were slightly lengthened in the IOR paradigm for working memory for the word-related task to be inserted.”

3. The authors only propose to perform a t-test on reading times. I believe an additional analysis should also be reported that takes condition as a fixed effect and item as a random effect in a mixed effects model. Given the uncertainty regarding reading times for the French words, it is important to make sure that the results are not confounded by some outlier words. Reading times during the refreshing phase could also be analyzed to provide a general baseline of spoken times for the words.

To be sure we are responding correctly to the question, we would like to point out that the t-test we plan to perform is done on the response time (= latency to the onset of reading the word), as was done by Johnson et al., and not on the reading time of words (= average time to read aloud). Regarding reading times, in the method section of the previous manuscript, it was already explained stated that while Johnson and colleagues took into account the reading times of the words to create word lists, we did not have that information for our French words and could hence not balance these reading times. The reading times of the words are hence never used/taken into account in this experiment. Despite this difference for the creation of the wordlists, this does not seem to have any consequences for the analysis. In Johnson’s experiment and ours, analyses are done on the response times. We think that the importance of reading times depends on the outcome of our replication (which we cannot share at this stage). If we replicate the IOR effect in our study, it is unlikely that the effect is caused by some outlier words (either in our study or in the previous studies by Johnson and colleagues). However, if we do not replicate the IOR effect in our study, we plan to examine the reason for the non-replication and the proposed analysis by the reviewer would be a good starting point. Accordingly, we have added on p. 13: **“Additional exploratory analyses.** *“In case our study does not replicate the inhibition-of-return-like effect observed by Johnson et al.¹, a series of additional exploratory analysis will be run to find out where this discrepancy comes from, including analyses that explore potential differences between the English and French stimuli.”*

Regarding the proposed analysis on the refreshing timings: As explained in response to Point 6 of R2, the policy of our local ethical committee is to allow only as much recordings as is necessary for the study design. As the original manuscript did not report any analysis on the refreshing times and we deemed it not necessary to record these refreshing times.

4. The authors should consider whether baseline reading times differ between their study and the ones previously reported in the literature that they are aiming to replicate. This may deserve a brief discussion.

We refer to the previous point regarding the terminology “reading time” (= average time to read aloud). We have no information regarding the reading times of the words used in our sample. Regarding the response times, we do indeed intend to make a comparison in the discussion section between the response times in the present experiment, and the studies by Johnson (2013 and 2020) using a similar experimental design.

5. I was not sure about this exclusion of the trials with reading times below 150 ms. Wouldn't this affect only previously refreshed trials? because in neutral and unrefreshed trials, there would be no way a delayed speech onset from the refreshing phase could lead to a correct response for analysis. To alleviate these concerns, an analysis including and excluding these trials should be provided, as well as specific information about how many trials were excluded because of this in each design cell.

Excluding trials with **response times** below 150 ms would in this experiment not differently affect refreshed, unrefreshed or neutral trials. A delayed speech onset from the refreshing phase could lead to a correct response of the probe word in any of the trials, as it is the experimenter that takes notes whether the probe word was read aloud correctly (as well as whether the refreshed word was correctly said aloud). If both words are said/read aloud correctly, the experimenter ticks the trial as correct. The human ear can however not discriminate in which time window the end of the refreshed word was said aloud. (Recordings of the trials could eliminate this uncertainty, yet as stated above we did not ask for ethical approval for these recordings, we deemed it sufficient to exclude trials with sound recording before 150 ms, and that is also what had been preregistered.). Additionally, we can say that this kind of situations was rather rare, both in our study and in the original study (see also information given above by the author of the original article, here Reviewer 2). In the Phase 2 of the manuscript submission, we will give specific information on how many trials were excluded and we can indeed split this information regarding the different trial types to ascertain these exclusions are not linked to one of the trial types in particular.

Appendix B

Dear Professor Chambers,

The manuscript we are submitting today is the stage 2 submission of Manuscript RSOS-210254.R1, after having received in principle acceptance for the stage 1 manuscript on 06.04.2021. As a reminder, our study aimed at replicating inhibition of return in working memory.

We have registered the approved stage 1 protocol on OSF (<https://osf.io/59rjk>), and the data as well as the study materials are available on OSF as well (<https://osf.io/3k6rv/>). We have left the stage 1 submission of the manuscript, including abstract, introduction and method section as they were, except for the correction of some typos and some information about the exclusion rates that was foreseen to be filled in. These modifications from the stage 1 IPA are all highlighted in yellow in this stage 2 manuscript, for which we have now completed the results, discussion and conclusion sections.

We hope that this Stage 2 registered replication study is satisfying the requirements for publication in the Royal Society Open Science journal.

Kind Regards,

Naomi Langerock, Giuliana Sposito, Caro Hautekiet & Evie Vergauwe

Appendix C

Regards,

Here is my review of the stage 2 manuscript by Langerock et al.

Background/Summary:

- In a 2013 study, Johnson et al. published a 4-experiment paper which shows that participants are slower to respond to stimuli which were 'refreshed' (a sort-of 'cued reflection') immediately before, compared to stimuli which were not refreshed, but were previously presented. An analogy was made with IOR, an attentional effect in which participants are slower to respond to/fixate on previously-disengaged locations.
- Some participants also completed a memory task, which showed that refreshed-probe trials have improved recall, adding a layer of intrigue to this effect.
- Because this runs contrary to the understanding of how refreshing increases activation, instead of suppresses (re: Johnson), and because this has not been supported by other research (to their knowledge), the current authors look to replicate one of the experiments presented.
- The results clearly indicate a replication using Bayes analysis, and the authors do a great job explaining these outcomes and their process of running participants until sufficient evidence is accumulated.
- The discussion compares the similarities in the effect sizes, and some minor statistical and methodological differences to the original paper and indicate that this is strong support for the effects reported in the original paper in the face of other studies.

Disposition

This was a clear replication of the original study by Johnson et al., with the authors doing an excellent job outlining all the important components and transparently going through all the checks required. I only have some minor comments that should be considered, but otherwise strongly recommend it for publication.

Comments and Concerns

- The results and discussion sections are written very clearly. It was easy and enjoyable to read.
- I was unable to access the data on OSF.
- I appreciate your statistical choices, and how they lend to generating appropriate power to detect the effects. Where the BF #s reported range quite a bit in size (from almost 800k to 31k to 2.54), and where this is a **relatively** novel method for analyzing stats, it would be great to have a reference point to indicate what these larger sizes mean (for instance 'anything above X is considered very strong evidence'). You do this with your refreshed probes main analysis, indicating under 3 is inconclusive, which is great, but maybe a quick comment on effects in the other direction will help unfamiliar readers better interpret these stats, and understand whether there is a meaningful difference between a BF of 800k and 31k.
- I would like to see the authors speak a bit more as to why they think there was this divergence in results for other research in the discussion section (page 14, line 19).

- With consideration to the statement made on page 14, line 19, I feel like calling this effect robust and reliable in the conclusion is potentially a bit overstated. The effect has certainly been replicated successfully in this case when the methodology was identical, but in the presence of other conflicting results when using other methodologies, it's not exactly clear how robust this working memory IOR effect really is. That being said, it certainly is promising and will surely lead to future research to help parse this out.

Appendix D

Dear Professor Chambers,

We thank you and the reviewers for the comments and suggestions made on this stage 2 submission of our manuscript “Inhibition-of-return-like effects in working memory? A preregistered replication study of Johnson et al. (2013)”; RSOS-210254.R2.

We have responded to the remaining comments and made the adaptations in the main text in yellow. We hope that this revision meets the standards for publication in the Royal Society Open Science Journal.

Kind Regards,

Naomi Langerock, Caro Hautekiet, Giuliana Sposito, & Evie Vergauwe

Associate Editor's comments (Professor Chris Chambers):

The three original reviewers from Stage 1 returned to assess the Stage 2 manuscript, and their evaluations are helpful and broadly positive. The reviewers note a range of areas that would benefit from clarification or elaboration, which I won't summarise here, but the main issue to address is the point raised by Reviewer 2 concerning the accuracy of the results. All of the reviewers note that the data are not (yet) available on the OSF and Reviewer 2 questions whether there may be a reporting error in the Bayesian analyses. Please attend carefully to this point and check that all reporting is accurate, and I would ask also that full open data and code are made available with the next revision so that, if necessary, we can engage the reviewers for any rechecking.

Regarding the general remark, raised by each reviewer, that the data and material were not publicly available, we sincerely apologize for this inconvenience. It was our intention to make the data as well as the materials publicly available. Although the main folder on OSF was made public, the subfolders, containing the data and materials, had not explicitly been set to “publicly available” and had, as such, remained private. This error has now been corrected.

Reviewers' Comments to Author:

Reviewer: 1

Comments to the Author(s)

Regards,

Here is my review of the stage 2 manuscript by Langerock et al.

Background/Summary:

- In a 2013 study, Johnson et al. published a 4-experiment paper which shows that participants are slower to respond to stimuli which were 'refreshed' (a sort-of 'cued reflection') immediately before, compared to stimuli which were not refreshed, but were previously presented. An analogy was made with IOR, an attentional effect in which participants are slower to respond to/fixate on previously-disengaged locations.
- Some participants also completed a memory task, which showed that refreshed-probe trials have improved recall, adding a layer of intrigue to this effect.
- Because this runs contrary to the understanding of how refreshing increases activation, instead of suppresses (re: Johnson), and because this has not been supported by other research (to their knowledge), the current authors look to replicate one of the experiments presented.
- The results clearly indicate a replication using Bayes analysis, and the authors do a great job explaining these outcomes and their process of running participants until sufficient evidence is accumulated.
- The discussion compares the similarities in the effect sizes, and some minor statistical and methodological differences to the original paper and indicate that this is strong support for the effects reported in the original paper in the face of other studies.

Disposition

This was a clear replication of the original study by Johnson et al., with the authors doing an excellent job outlining all the important components and transparently going through all the checks required. I only have some minor comments that should be considered, but otherwise strongly recommend it for publication.

Comments and Concerns

- The results and discussion sections are written very clearly. It was easy and enjoyable to read.
- I was unable to access the data on OSF.

This issue has been resolved. Thank you for drawing our attention towards this.

- I appreciate your statistical choices, and how they lend to generating appropriate power to detect the effects. Where the BF #s reported range quite a bit in size (from almost 800k to 31k to 2.54), and where this is a relatively novel method for analyzing stats, it would be great to have a reference point to indicate what these larger sizes mean (for instance 'anything above X is considered very strong evidence'). You do this with your refreshed probes main analysis, indicating under 3 is inconclusive, which is great, but maybe a quick comment on effects in the other direction will help unfamiliar readers better interpret these stats, and understand whether there is a meaningful difference between a BF of 800k and 31k.

We added on page 10, after reporting the Bayes factor of 799082 and 31326 (following Schönbrodt & Wagenmakers, 2018): *Generally, Bayes factors exceeding 10 are considered strong evidence, and exceeding 100, as is the case here, as extreme evidence*²⁶.

Further on, after reporting the Bayes factor of 2.54 it was already mentioned that “Bayes factors below three are generally considered inconclusive²⁶.”

- I would like to see the authors speak a bit more as to why they think there was this divergence in results for other research in the discussion section (page 14, line 19).

At the moment, a large number of hypotheses are possible, and we are doing some follow-up studies in our lab to test some of these. Regarding the manuscript, we have completed the last sentences before the discussion as follows:

At the moment, several hypotheses as to why these results diverge could be proposed, based on methodological differences between these different studies. We will at this point refrain from speculating about these hypotheses and leave it up to future studies to clarify what is at the origin of this divergence and how these two effects (inhibition and facilitation) can be integrated into a comprehensive account of refreshing.

- With consideration to the statement made on page 14, line 19, I feel like calling this effect robust and reliable in the conclusion is potentially a bit overstated. The effect has certainly been replicated successfully in this case when the methodology was identical, but in the presence of other conflicting results when using other methodologies, it's not exactly clear how robust this working memory IOR effect really is. That being said, it certainly is promising and will surely lead to future research to help parse this out.

We agree with reviewer 1 that it may be misleading to call the effect robust and reliable as it has been replicated within this particular experimental design but apparently not when using other designs, or when introducing small deviations. We have adapted the first sentence of the conclusion:

The current replication shows that the inhibition-of-return-like effect in working memory can reliably be observed across different labs and languages when using this exact same paradigm.

Reviewer: 2

Comments to the Author(s)

Hello again. It is one of your reviewers from last time, Matt Johnson, still with the help of my Ph.D. students Evan Lintz and Zach Cole. I think we were clear before that we are enthusiastic about this line of work generally, and we offered most of what we have to say on the previous round of reviews before results were added in, so we'll just jump right in and try to be brief.

Major comments:

Only one major one here. This may not end up being an issue at all, but we wanted to make sure. If we are understanding everything correctly, there seems to be a curiously large jump in the Bayes factor for the primary hypothesis when going from N=28 (2.54) to N=33 (28). Of course almost anything is statistically possible in the unpredictable world of human subjects data, but it seemed intuitively unlikely

so we dug in a bit further. We would have tried working with the actual data, but it appears it is not yet available at the OSF link included in the paper? (This is perfectly fine if the authors want to wait to make it available until after acceptance... we just wanted to note that we tried that option first.) So next we ran some simulations to generate a few datasets that had a BF of around 2.5 at N=28, and then tried to assess what the likelihood would be of observing a $BF \geq 28$ after adding just five more subjects drawn randomly from the same underlying distribution. Naturally a number of assumptions go into such simulations (e.g. that values are normally distributed and so on), so it is hard to know how well they approximate reality... but in any case, it seemed like adding five subjects and seeing the BF jump from ~ 2.5 to ~ 28 was exceedingly rare (on the order of 1 out of every 10,000 simulations or so). Anyway, this is a long way of saying... is it possible that the BF of 28 at N=33 could be a typo or some other kind of error? If you are confident in those values, then that is great and we're sorry to waste your time... but the difference was striking enough that hopefully you can understand our surprise.

The Bayes factor after having tested 33 participants is reported accurately and the result of this one-sided t-test corresponds thus indeed to $BF_{10} = 28$. A simple data-check could have resolved this issue, and we once again apologize for the non-availability of the data (the issue has been resolved in the meantime, thank you for noticing). We hereby provide the graph showing the sequential analysis for the main hypothesis-testing, which displays a gradual accumulation of evidence towards the IOR-effect while adding more data.

Minor comments

(Note that there are several ways to number pages. The authors appear to have originally numbered odd-numbered pages. It looks like the submission system added its own numbering system to the proof. And the page within the PDF file is also different. For example, in our proof, the page the authors originally numbered as 7 is printed in the RSOS header as "Page 9 of 18" and it is the 10th page in the PDF file. We will use the page numbers in the RSOS header in our notes below, e.g. "Page 9 of 18")

page 4, line 55: Typo... the name should be Kiyonaga

corrected

page 5, line 11: It could be confusing to say words were presented "vertically," which sort of sounds like the individual letters in the words were printed in a vertical line, rather than the intended meaning (that both words were printed normally but one was vertically above the other). So you may wish to rephrase for clarity.

Adapted to: two words displayed on-screen one above the other

page 7, line 54: The usage of the word "only" is a bit out of place here, since $2000 > 1600$. We recommend just removing the word "only" on this line.

corrected

page 12, "Additional exploratory analyses" section: This section still is written as if it is in a manuscript where the results are not yet known, so it sounds a bit odd to say things like "In case our study does not replicate... the effect..." You could potentially take out this section entirely, since it ultimately appears to have been irrelevant... or if you want to leave it in, it might be good to rephrase it a bit to clarify that those steps WOULD have been taken if the replication did not occur. (This might just be a simple issue of verb tenses/moods.)

We have moved this section to a footnote and additionally changed the phrasing to "would". As this section was part of the IPA, it would be most fair to leave it in, while it is true at the same time that this section is now less relevant within the manuscript.

References -- the Lintz & Johnson paper is now published, not merely forthcoming, so you can update this reference. (Lintz EN, Johnson MR. 2021. Refreshing and removing items in working memory: Different approaches to equivalent processes? *Cognition* 211: article 104655.)

This reference has been updated, thank you for indicating this.

We hope these comments were helpful, and we remain enthusiastic about this line of work!

Best wishes,
Matt, Evan, and Zach

Reviewer: 3

Comments to the Author(s)

In general, the paper is well-written and the analysis and results were clear cut. The only pending issue in view was that the data and materials were not really accessible as required by the journal. There was no data available in the OSF page indicated. It was completely blank. I was confused if this was a feature required by the journal - but then, they ask us to evaluate whether the

data-posting was clear, which it is not possible as it is.

The issue about the availability of the data has been resolved, thank you for noticing.